# Analysis of streamflow response to land use land cover changes using satellite data and hydrological modelling: case study of Dinder and Rahad tributaries of the Blue Nile (Ethiopia/Sudan)

Khalid Hassaballah[1, 2, 3], Yasir Mohamed[1, 2, 3], Stefan Uhlenbrook[1, 2, 4], Khalid Biro[3]

[1]UNESCO-IHE Institute for Water Education, P.O. Box 3015, 2601DA Delft, The Netherlands

[2]Delft University of Technology, P.O. Box 5048, 2600 GA Delft, The Netherlands

[3]The Hydraulics Research Center, P.O. Box 318, Wad Medani, Sudan

[4]Unesco Italia, UN World Water Assessment Programme

*Correspondence to*: Khalid Hassaballah (k.hassaballah@yahoo.com)

**Abstract.** Understanding the land use and land cover changes (LULCC) and its implication on surface hydrology of the Dinder and Rahad basins (D&R) approximately 77,504 km$^2$ is vital for the management and utilization of water resources in the basins. Although there are many studies on LULCC in the Blue Nile basin, specific studies on LULCC in the D&R are still missing. Hence, its impact on streamflow is unknown. The objective of this paper is to understand the LULCC in the Dinder and Rahad and its implications on streamflow response using satellite data and hydrological modelling. The hydrological model has been derived by different sets of LULC maps from 1972, 1986, 1998 and 2011. Catchment topography, land cover and soil maps, are derived from satellite images and serve to estimate model parameters. Results of LULCC detection between 1972 and 2011 indicate a significant decrease of woodland and an increase of cropland. Woodland decreased from 42% to 14% and from 35% to 14% for Dinder and Rahad respectively. Cropland increased from 14% to 47% and from 18% to 68% in Dinder and Rahad respectively. The model results indicate that streamflow is affected by LULCC in both the Dinder and the Rahad Rivers. The effect of LULCC on streamflow is significant during 1986 and 2011. This could be attributed to the severe drought during mid 1980s and the recent large expansion in cropland.

**Keywords:** land use land cover, PCRaster, hydrological modelling, streamflow response, WFlow model, Dinder Rahad

## 1    Introduction

Streamflow is an important hydrological variable needed for water resources planning and management, and for ecosystem conservations. The rainfall runoff process over the upper Dinder and Rahad basins (D&R) is complex, non-linear, and exhibits temporal and spatial variability (Hassaballah et.al. 2016). To manage water resources effectively at a local level, decision makers need to understand how human activities and climate change may impact local streamflow. Therefore, it is necessary to understand the hydrological processes in the run-off generated catchments, and the possible interlinkages of land use and land cover changes with catchment runoff. For this reason, we used satellite data and hydrological modelling to analyze the land use and land cover changes (LULCC) and its impacts on streamflow response in the D&R.

The D&R generate around 7% of the Blue Nile basin's annual flow. The Rahad River supplies water to the Rahad Irrigation Scheme (100,000 ha), while the Dinder River supplies water to the diverse ecosystem of the Dinder National Park (DNP). The DNP (10,291 km$^2$) is a vital ecological area in the arid and semi-arid Sudan-Saharan region.

The Dinder and Rahad Rivers have experienced significant changes in floodplain hydrology during recent years, claimed to be caused by land use land cover changes in the upstream catchment. The floodplain hydrology defines the seasonal wetlands (Mayas) which are the only source of water in the DNP during the dry season (8 months). The hydrology of the mayas has large implications on the ecosystem of the DNP. A detailed description of the mayas wetlands can be found in Hassaballah et al. (2016).

LULCC was identified as a key research priority with multi-directional impacts on both human and natural systems (Turner II et al. 2007). Many studies highlighted the impacts of LULCC on hydrology (DeFries and Eshleman 2004; Uhlenbrook 2007), on ecosystem services (DeFries and Bounoua 2004; Metzger et al. 2006; Polasky et al. 2011) and on biodiversity (Hansen et al. 2004; Hemmavanh et al. 2010).

LULCC is a widespread observable phenomenon in the Ethiopian highlands as pointed by (Zeleke and Hurni 2001; Bewket and Sterk 2005; Hurni et al. 2005; Teferi et al. 2013;). These studies have pointed out different types and rates of LULCC in different parts of the Ethiopian highlands over different time periods and reported that the expansion of croplands associated with a decrease in woodlands have been the general forms of transitions.

Recently, Gumindoga et al. (2014) assessed the effect of land cover changes on streamflow in the Upper Gilgel Abbay river basin in northwestern Ethiopia. Their results showed significant land cover changes where cropland has changed from 30% of the catchment in 1973 to 40% in 1986 and 62% in 2001. The study attributed these changes to the increase in population, which increased the demands for agricultural land. The study has also pointed that farmers in the area are commonly clearing forests to create croplands, and the resulting effect was the decrease in forest land from 52% in 1973 to 33% in 1986 and 17% in 2001. Since the Upper Blue Nile basin is neighboring the D&R, one may expect some similarities of catchment characteristics, though differences cannot be excluded. These transitions have contributed to the high rate of soil erosion and land degradation in the Ethiopian plateau (Bewket and Teferi 2009). Understanding the impacts of LULCC on hydrology, and incorporating this understanding into the emerging focus on LULCC science are the most important needs for the future (Turner et al. 2003).

Many models have been developed to simulate impacts of LULCC on streamflow. These can be categorized as an empirical black box, conceptual, and physically based distributed models. Each type of these three models has its own advantages and limitations. Several situations in practice demand the use of simple tools such as the linear system models or black box models. Nevertheless, these simpler models usually fail to mimic the non-linear dynamics, which are essential in the rainfall-runoff transformation process. Therefore, the development of a dynamic modelling language within a GIS framework such as PCRaster is a further important stage that allows complex models, like the WFlow rainfall-runoff model, to be implemented making use of globally available spatial data sets. The PCRaster programming language is an environmental modelling language to build dynamic spatial environmental models (Bates and De Roo 2000; Karssenberg 2002; Uhlenbrook et al. 2004).  Such spatially distributed models also have the potential to help in answering questions of policymakers about

the impact of spatial changes (e.g. impacts of LULCC on streamflow dynamic). It has been shown that a variety of probable LULCC impacts on hydrologic processes in the D&R are likely to happen. Therefore, the objective of this study is to understand the LULCC in the D&R and its impacts on streamflow response using satellite data, GIS and remote sensing, and hydrological modelling. The WFlow distributed hydrological model (Schellekens, 2011) is used simulate the processes. In addition, understanding the level to which the streamflow has altered is critical for developing an effective management plan for ecosystem restoration and conservation. Thus, the Indicators of Hydrological Alteration (IHA) approach proposed by Richter et al. (1996), was then applied to analyze the streamflow characteristics likely to affect the ecological processes in the D&R including: flow magnitude, timing, and rate of change of flow.

## 2    Study area

The Dinder and the Rahad are the lower sub-basins of the Blue Nile River basin located between longitude 33°30' E and 37°30' E and latitude 11°00' N and 15°00' N (Fig. 1). The Blue Nile basin collects flows of eight major tributaries in Ethiopia besides the two main tributaries in Sudan: the Dinder and the Rahad Rivers. Both tributaries derive their water from the runoff of the Ethiopian highlands approximately 30 km west of Lake Tana (Hurst et al. 1959). Their catchments areas are about 34,964 and 42,540 $km^2$ for the Dinder and the Rahad, respectively, giving a total area of about 77,504 $km^2$. The catchment has varied topography with elevation ranges between about 384 m at the catchment outlet and up to 2731 m at the Ethiopian plateau. The D&R have a complex hydrology, with varying climate, topography, soil, vegetation and geology (Hassaballah et al. 2016). The annual average flow is about 2.797 x $10^9$ and 1.102 x $10^9$ $m^3$/year for the Dinder and the Rahad, respectively.

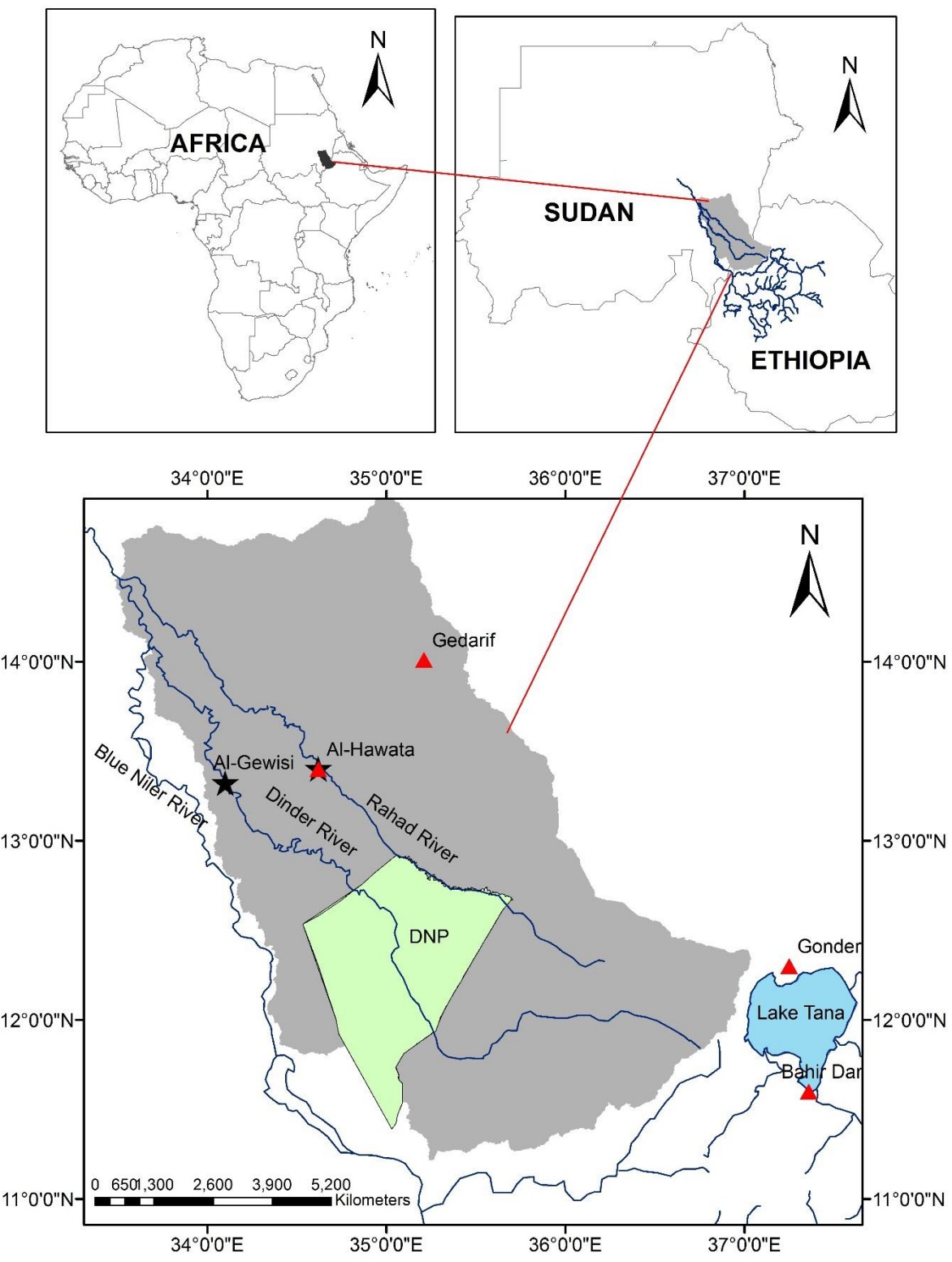

Figure 1: Location map of the Dinder and Rahad basins and the DNP. The two black stars show the locations of the hydrological stations (Al-Gewisi and Al-Hawata), and the red triangles show the locations of the rain gauges.

## 3    Data and Methods

Limited data is available for simulating the hydrology of the D&R. To fill this data gap, use has been made of globally available free datasets. The datasets which have been used to run the WFlow model are divided into two datasets; static data and dynamic data.

### 3.1    Input data

#### 3.1.1    Static data

The static data contain maps that do not change over time. It includes maps of the catchment delineation, Digital Elevation Map (DEM), gauging points, land use, local drainage direction (*ldd*), outlets and rivers. These maps were created with a pre-prepare processes of the WFlow hydrologic model.

The catchment boundary has been delineated based on a 90 m x 90 m digital elevation map (DEM) of the NASA Shuttle Radar Topographic Mission (SRTM) obtained from the Consortium for Spatial Information (CGIAR_CSI) website (http://srtm.csi.cgiar.org).

Multi-temporal Landsat data for the years 1972, 1986, 1998 and 2011 were obtained free of charge from the internet site of the United States Geological Survey (USGS) (source: http://glovis.usgs.gov/). All images were geometrically corrected into the Universal Transverse Mercator (UTM) coordinate system (Zone-36N).

The soil map was obtained free of charge from the Food and Agriculture Organization (FAO) Harmonized World Soil Database (HWSD). The original catchment boundary layer provided 44 Soil Mapping Units (SMU) classes. These classes have been reclassified into 8 dominant soil group (DSG) categories, based on the DSG of each soil mapping unit code. This was necessary to reduce the model complexity. The WFlow soil model requires estimates of 8 parameters per soil type, which means 352 parameters if it is for 44 soil types. Therefore, reclassification of soil map into 8 dominant soil groups reduces the number of estimated parameters to 64. The categories are: vertisols 71%, luvisols 9%, nitisols 8%, leptosols 5%, cambisols 4%, alisols 2% and fluvisols 1%. The map was then projected to WGS-84-UTM -zone-36N and resampled to a horizontal resolution of 500 m.

#### 3.1.2    Satellite based rainfall and evapotranspiration data

The dynamic data contain maps that change over time. It includes daily maps of the precipitation and evapotranspiration. These maps were created with a pre-prepare step1 and step2 of WFlow model. In this study, three open-access satellites-based rainfall estimates (SBRE) products were compared based on their runoff performance at Al-Gewisi and Al-Hawata stations the outlets of the Dinder and Rahad basins, respectively. The best product was then used to run the WFlow model using different LULC maps. The SBRE and the evapotranspiration products used in this study are: Rainfall Estimates (RFE 2.0), potential evapotranspiration (PET), Tropical Rainfall Measuring Mission (TRMM) and Climate Hazards Group InfraRed Precipitation with Stations (CHIRPS).

The RFE 2.0 and the PET data were obtained from the Famine Early Warning System Network (FEWS NET). The horizontal resolution is 0.1 degree (11.0 km) for the RFE and 1.0 degree (110 km) for PET. This data is available on a daily basis from 2001 to near real-time period of record. More description can be found at http://earlywarning.usgs.gov/adds/downloads/.

The TRMM is a joint space mission between NASA and the Japan Aerospace Exploration Agency (JAXA) launched in 1997. The TRMM satellite rainfall measuring instruments include the Precipitation Radar (PR), TRMM Microwave Image (TMI), a nine-channel passive microwave radiometer, a Visible and Infrared Scanner (VIRS), and five-channel visible/infrared radiometer (Huffman and Bolvin 2013). In this study, we have used TRMM 3B42v7 which has a spatial resolution of 0.25° and a temporal resolution of 3 hours. More information can be found at (www.trmm.gsfc.nasa.gov).

The CHIRPS data were developed by the Climate Hazards Group (CHG) and scientists at the U.S. Geological Survey Earth Resources Observation and Science Center. This product is a new quasi-global precipitation with daily to seasonal time scales, a 0.05° resolution, and 1981 to near real-time period of record. The CHIRPS uses the monthly Climate Hazards Precipitation Climatology (CHPClim), the InfraRed (IR) sensors from the Group on Earth Observations (GEO) satellites, the TRMM 3B42 product, and the ground precipitation observations. More information about CHIRPS data can be found in Funk et al. (2014). A summary of all precipitation and evapotranspiration satellite products was provided in Table 1. All maps were projected into WGS-84-UTM-zone 36N (meters), clipped to catchment extent, and then resampled to a resolution of 500 m.

Table 1: Summary of the different precipitation and evapotranspiration satellite products

| Product | Developer | Spatial resolution | Covering area | Temporal resolution | Time span | Ground measurement |
|---------|-----------|--------------------|---------------|---------------------|-----------|--------------------|
| TRMM 3B42v7 | NASA, JAXA | 0.25° | 0°E-360°E/50°N-50°S | 3 Hourly | Jan 1998 - present | Yes |
| RFE 2.0 | NOAA (CPC) | 0.1° | 20°E-55°E/40°N-40°S | 6 Hourly | Jan 2001 - present | Yes |
| CHIRPS v2.0 | CHG | 0.05° | 0°E-360°E/50°N-50°S | Daily | Jan 1981 - present | Yes |
| PET | NOAA (CPC) | 1.0° | 20°E-55°E/40°N-40°S | 6 Hourly | Jan 2001 - present | Yes |

### 3.1.3    Observed hydrological streamflow

Daily streamflow data at Al-Gewisi station on the Dinder River and at Al-Hawata station on the Rahad River for the period (2001-2012) were obtained from the Ministry of Water Resources, Irrigation and Electricity-Sudan. This data is mainly used for calibration and validation of the WFlow hydrological model.

### 3.2    LULC classification and change detection

LULC images were selected in the same season to minimize the influence of seasonal variations on the classification result. All acquired images had less than 10% cloud cover. However, in order to cover the entire study area, more than 8 scenes of the satellite data were processed (Table 2). Subsequently, all images were mosaicked and resampled to a pixel size of 30m x 30m. The classification results of the historical images 1972, 1986 and 1998 were validated through visual interpretation of the unclassified satellite images and supported by in-depth interview of local elders. The classification of the 2011 image was validated by ground survey during a field visits throughout the study area during the period between 2011 and 2013 assuming no significant change during this period. A Global Positioning System (GPS) device was used to obtain exact location point data for each LULC class included in the classification scheme and for the creation of training sites and for signature generations as well. Moreover, field notes, site descriptions, and terrestrial photographs were taken to relate the site location to scene features. A total of (120) training areas were selected based on image interpretation keys, established during the field survey and from interviews with the local people. This later step was used as a crosscheck validation for the visual interpretation performed to the historical images. A supervised Maximum Likelihood Classification (MLC) technique was independently employed to the individual images. MLC is the most common supervised classification method used with remote sensing image data (Ellis et al. 2010; Pradhan and Suleiman 2009). The derivation of MLC is generally acceptable for remote sensing applications and is used widely (Richards et al. 2006).

The accuracy assessment of the classified images was based on the visual interpretation of the unclassified satellite images (Biro et al. 2013). However, the visual interpretation was conducted by an independent analyst not involved in the

classification. The stratified random sampling design, where the number of points was stratified to the LULC types, was adopted in order to reduce bias (Mundia and Aniya 2006). Accordingly, error matrices as cross-tabulations of the classified data vs. the reference data were used to evaluate the classification accuracy. The overall accuracy, the user's and producer's accuracies, and the Kappa statistic values were then derived from the error matrices.

5    Multi-date Post-Classification Comparison (PCC) change detection method described by Yuan et al. (2005) was used to determine the LULCC in three intervals: 1972–1986, 1986–1998 and 1998–2011. PCC is a quantitative technique that involves an independent classification of separate images from different dates for the same geographic location, followed by a comparison of the corresponding pixels (thematic labels) in order to identify and quantify areas of change (Al Fugara et al. 2009; Jensen 2004). It is the most commonly used method of LULCC detection mapping (Kamusoko and Aniya 2009).

Table 2: Description of used satellite images.

| Acquisition date | Satellite | Number of scenes | Spectral bands | Spatial resolution |
|---|---|---|---|---|
| 04 Nov. & 11 Dec. 1972 | Landsat MSS | 9 | 1 – 4 bands | 60 m |
| 12 Nov. & 26 Nov. 1986 | Landsat TM | 9 | 1 – 6 bands | 30 m |
| 27 Nov. & 13 Dec. 1998 | Landsat TM | 8 | 1 – 6 bands | 30 m |
| 07 Nov. & 10 Dec. 2011 | Landsat TM | 8 | 1 – 6 bands | 30 m |

MSS, multispectral scanner; TM, thematic mapper

### 3.3    Description of the WFlow hydrological Model

In order to assess the impacts of LULCC on the streamflow dynamic, the WFlow distributed hydrological model (Schellekens, 2011) is forced using SBRE. The WFlow is a state-of-the-art open source distributed catchment model. The model is part of the Deltares OpenStreams project *(http://www.openstreams.nl)*. The model is derived from the CQFLOW model (Kohler et al., 2006). It is a hydrological model platform that includes two models: the WFlow_sbm model described by Vertessy and Elsenbeer (1999) derived from the TIOPG_SBM soil concept, and the WFlow_hbv model (distributed version of the HBV model). The model directly appeals to the need within the hydrological and geomorphologic sciences community to effectively

use spatial datasets e.g. digital elevation models, land use maps, dynamic satellite data for rapid and adequate modelling of river basins with limited data availability. The model is programmed in PCRaster GIS dynamic language (Deursen 1995).

In this study, the WFlow_sbm PCRaster-based distributed hydrological model which makes use of the Gash and the TOPOG_SBM models was used. The model requires less calibration and maximizes the use of available spatial data that makes

it a suitable model for this study. Step one of WFlow model was to delineate river network and the gauging points based on the DEM. Next, a land use and soil maps were added to the model and parameters were estimated based on physical characteristics of the soil and land use type. The rainfall interception was calculated using the Gash model (Gash 1979, 1995), while hydrologic processes that cause a runoff or overland flow were calculated using the TOPOG_SBM model. The WFlow uses potential evapotranspiration as an input data and derives the actual evaporation based on soil water content and vegetation

cover type. The analytical model of rainfall interception in the WFlow is based on Rutter's numerical model (Gash, 1979; Gash et al., 1995). The surface runoff is modelled using a kinematic wave routine. Combination of the total rainfall and evaporation under condition of saturated canopy is done for each rainfall storm to determine average values of precipitation and evaporation from the wet canopy. In case the soil surface is partially saturated, the rainfall that falls on the saturated area is directly added to the surface runoff component. The soil is represented by a simple bucket model that assumes an exponential

decay of the saturated conductivity with depth. Lateral subsurface flow is simulated using the Darcy equation. Soil depth is identified for different land use types and consequently scaled using the Topographic Wetness Index. Different parameters are assigned to each land cover type. These parameters include; rooting depth, leaf area index (LAI), ratio of evaporation from wet canopy to average rainfall ($E_w/R$), Albedo, Canopy Gap Fraction and Maximum Canopy Storage. All model parameters

are linked to the Wflow model through lookup tables. The lookup tables are used by the model to create input parameter maps. Each table consists of four columns. The first column is used to identify the land-use class, the second column indicates the sub catchment, the third column represents the soil type and the last column lists the assigned values based on the first three columns. The parameters are linked to land use, soil type or sub-catchment through lookup tables. Description of the Wflow model parameters is presented in Appendix B and the calibrated values for each parameter are presented in Appendix C. The WFlow_sbm interception and soil model's equations are presented in Appendix A." Further details of the Wflow model are also given at https://media.readthedocs.org/pdf/wflow/latest/wflow.pdf.

The model is fully distributed, which means that it makes the calculations for every grid cell of the basin. Each cell (500 m x 500 m) is seen as a bucket with a total depth divided to saturated and unsaturated stores (Fig. 2). The streamflow model results were then analyzed using the IHA approach described by Richter et al. (1996).

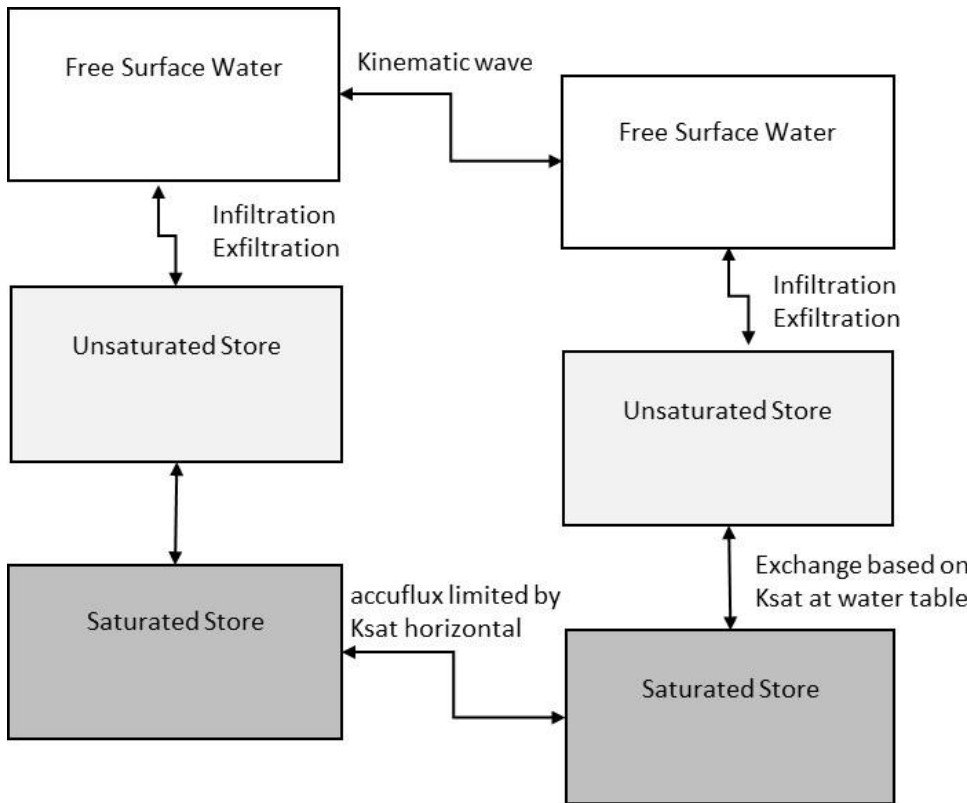

**Figure 2.** Schematization of the soil within the WFlow_sbm model
Source: http://WFlow.readthedocs.io/en/latest/WFlow_sbm.html#the-soil-model

### 3.3.1 Model calibration and validation

As with all hydrological models, calibration of the Dinder and Rahad hydrological model is needed for optimal performance. Since the hydrological data available for calibration start from 2001, the nearest land use (land use of 1998) was used in the calibration. The calibration procedure performed in two steps based on; firstly, initial values of all parameters were estimated based on the land use and the soil types. Secondly, by adjusting the model parameters and evaluate the results.

The performance of the model was assessed using measures of goodness of fit between the modeled and observed flow using the coefficient of determination ($R^2$) and the Nash–Sutcliffe efficiency (NSE), defined by Nash and Sutcliffe (1970). The

observed and the simulated flow of the Dinder and Rahad correlated well, except for few under-predictions and over-predictions of peak flows which can be explained in terms of inherent uncertainty in the model and the data. However, measures of performances for both calibration and verification runs fell within the acceptable ranges.

### 3.4    Indicators of hydrologic alterations (IHA)

The IHA approach was introduced by Richter et al. (1996). The approach used to assess river ecosystem management objectives defined based on a statistical representation of the most ecologically relevant hydrologic indicators. These indicators describe the essential characteristics of a river flow that have ecological implications. The IHA method computes 33 hydrologic parameters for each year. For analyzing the alteration between two periods, the IHA described in Richter et al. (1996) was applied using the IHA software developed by The Nature of Conservancy (2009).

The general approach is to define hydrologic parameters that characterized the intra-annual variation in the water system condition and then to use the analysis of variations in these parameters as a base for comparing hydrologic alterations of the system before and after the system has been altered by various human activities.

The IHA method has four steps: a) define the time series of the hydrologic variable (e.g. streamflow) for the two periods to be compared; b) calculate values for hydrologic parameters; c) compute intra-annual statistics; and d) calculate values of the IHA by comparing the intra-annual variation before and after the system has been altered and present the results as a percentage of deviation. For assessing hydrologic alteration in the Dinder and Rahad Rivers, the flows variations for both rivers have been characterized based on the variations in the streamflow characteristics between three periods (1972-1986), (1986-1998) and (1998-2011). Temporal variability of streamflow series was analyzed at Al-Gewisi station on the Dinder River and at Al-Hawata station on the Rahad River. A detailed description of IHA can be found in (Richter et al., 1996 and Poff et al., 1997).

### 4    Results and Discussion

### 4.1    LULC classification and change detection

The overall LULC classification accuracy levels for the four images ranged from 82% to 87%, with Kappa indices of agreement ranging from 77% to 83% (Table 3). The accuracy assessment is based on comparing reference data (class types at specific locations from ground information) to image classification results at the same locations. The overall accuracy of classification is the average value from all classes. The user's accuracy corresponds to errors of inclusion (commission errors), which represents the probability of a pixel classified into a given class actually represents that class on the ground (i.e. from the perspective of the user of the classified map). The producer's accuracy corresponds to errors of exclusion (omission errors), which represents how well reference pixels of the ground cover type are classified (i.e. from the perspective of the maker of the classified map). The commission errors occur when an area is included in an incorrect category, while the omission errors occur when an area is excluded from the category to which it belongs. Every error on the map is an omission from the correct class and a commission to an incorrect class (Congalton and Green 2008). The cross validation for the year 2011 land use was made using the reference data (120 points) collected with GPS instrument during the field survey (2011–2013). In addition, visual interpretation and historical information obtained from the local people about the land use types in the study area were used also as cross-check validations for old maps. Shrublands show lower user's and producer's accuracies compared to the other LULC classes. This is mainly due to the miss-classification of some shrub land into woodland, grassland and cropland. This accuracy is satisfactory for the study area considering the multi-temporal analysis of Landsat data and the visual interpretation adapted to image classification.

Table 3: Accuracy assessment (%) of LULC maps.

| LULC Classes | 1972 | | 1986 | | 1998 | | 2011 | |
|---|---|---|---|---|---|---|---|---|
| | Producer's | User's | Producer's | User's | Producer's | User's | Producer's | User's |
| Woodland | 88 | 89 | 89 | 90 | 89 | 90 | 91 | 93 |
| Cropland | 78 | 70 | 80 | 74 | 80 | 80 | 83 | 82 |
| Shrubland | 71 | 71 | 73 | 75 | 77 | 75 | 80 | 75 |
| Grassland | 80 | 88 | 83 | 88 | 85 | 88 | 86 | 89 |
| Bare Land | 82 | 76 | 82 | 78 | 82 | 78 | 82 | 85 |
| Water | 86 | 86 | 88 | 86 | 91 | 86 | 94 | 86 |
| Overall | 82 | | 84 | | 85 | | 87 | |
| Kappa | 77 | | 79 | | 81 | | 83 | |

Landsat image classification results for the years 1972, 1986, 1998 and 2011 are shown in Figure 3. The large extent of the catchment (77,504 km2), and the small-scale of the maps (i.e. 1: 4,500,000), may not allow distinction of different LULC change patterns by eye. Figure 4 which zoom into smaller area is an example to show multi-temporal changes in the LULC patterns. The zoomed areas in the red boxes showed in large-scale, provides more details of LULC patterns. This area is located downstream of the Rahad Irrigation Scheme of Sudan established in 1981. The waterlogging and woodland areas occurred in 1998 and 2011 resulted from the drainage water of the project accumulated over the years (i.e. clear example of LULC multi-temporal change over the Rahad basin). The lower maps show the Google Earth imagines of the large-scale area. Although these Google Earth imagines dates do not exactly match the ones of the satellite images, they show the part of the dried period in the study area and hence the complexity of the LULC patterns.

According to the produced LULC maps, it was found that woodland, shrubland and grassland were the dominant types of LULC classes for the years 1972, while for the year 1986 they were shrubland, grassland and cropland. The LULC map of 1998 illustrates that the predominant types of LULC classes were cropland and woodland, while they were cropland and shrubs in 2011.

LULCC in the D&R are assessed by image comparison. In general, the results showed that the dominant process is the large decrease of woodland and increase of cropland. This result was in agreement with that of Rientjes et al. (1979) and Gumindoga et al. (2014), who studied the changes in land cover, rainfall and streamflow in the neighboring catchment of the upper Gilgel Abbay in Ethiopia.

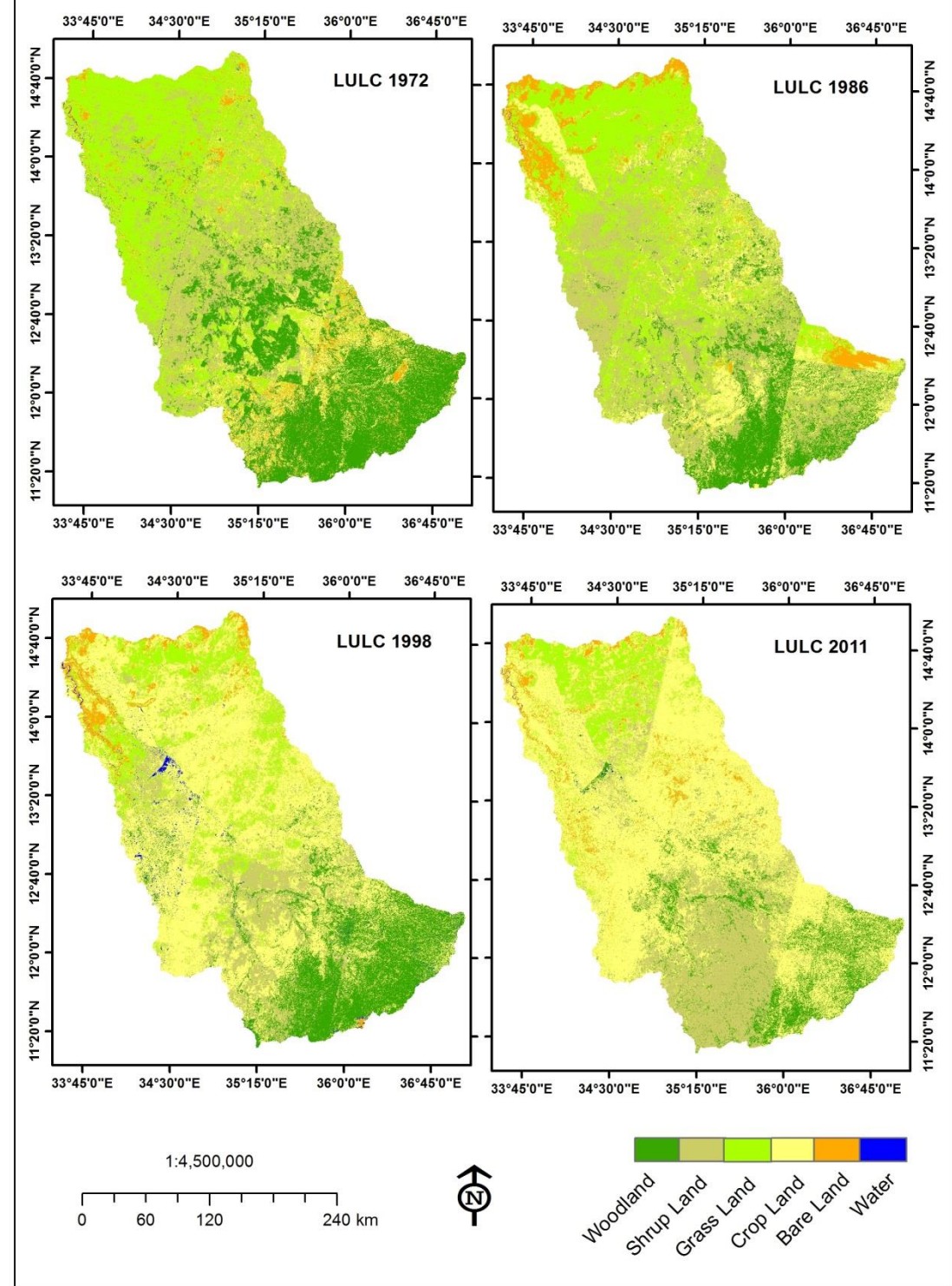

**Figure 3.** Classified LULC maps of the years 1972, 1986, 1998 and 2011.

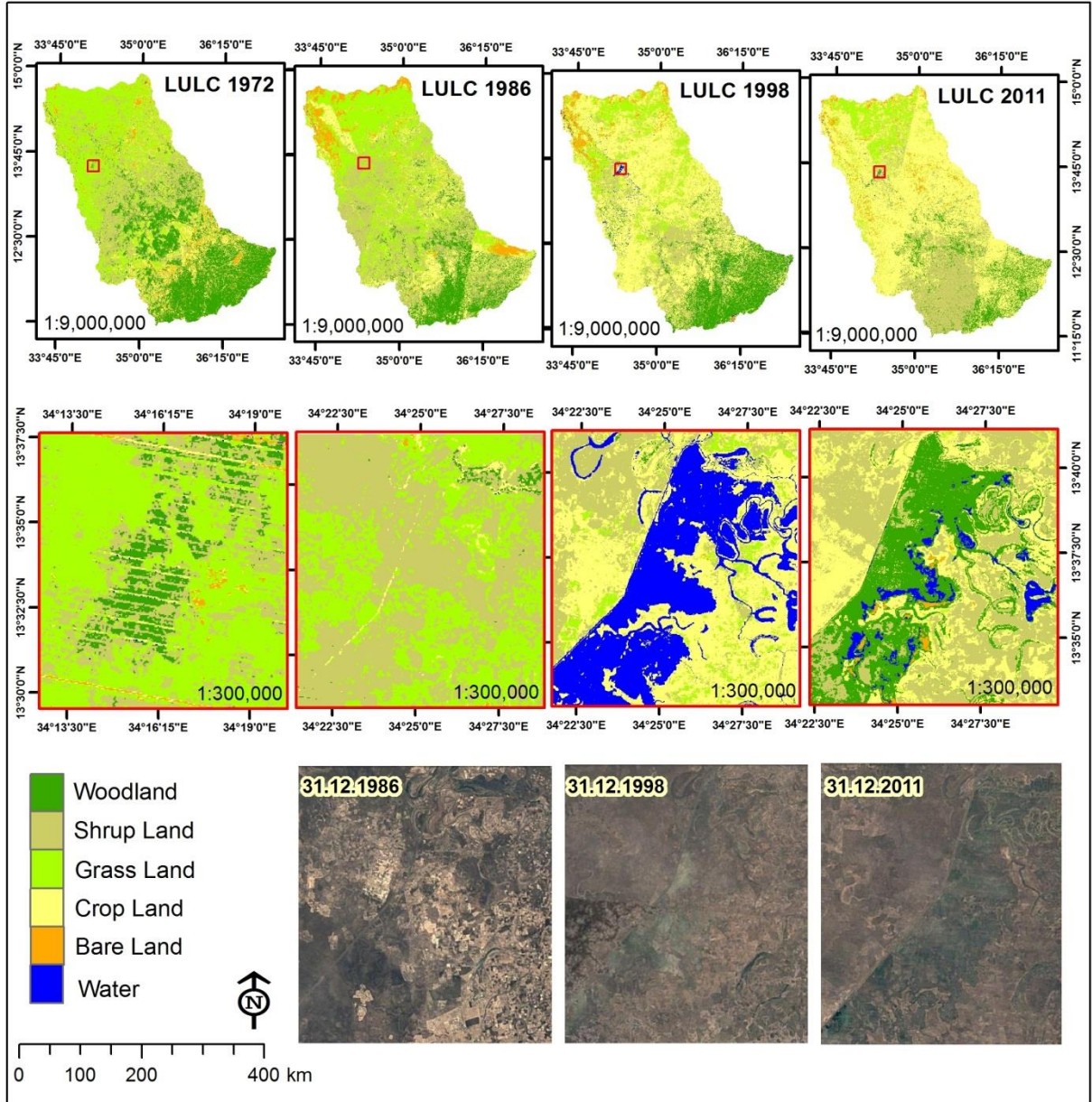

**Figure 4**: Classified LULC maps of the years 1972, 1986, 1998 and 2011. The areas in the red boxes showed in large-scale provide more details of the LULCC patterns.

Table 4 shows the percentages of LULCC classes in Dinder and Rahad basins that occurred in the period 1972 to 1986, 1986 to 1998, and 1998 to 2011. The decrease in the woodland area in 1986 is mainly attributed to the deforestation during the drought time in 1984 and 1985. As a result, the cropland was increased due to the development of new agricultural areas in both irrigated (i.e. Rahad Agricultural Scheme) and rain-fed sectors. The rapid expansion in the mechanized rain-fed agriculture led to a large increase in cropland during 1998 and 2011. These findings are in agreement with what have been reported by Marcotullio and Onishi (2008), and Biro et.al, (2013) from their similar studies conducted in the Ethiopian highlands and Gedarif region in eastern Sudan.

Table 4: Land cover changes (%) in Dinder and Rahad basins that occurred in the period 1972 to 1986, 1986 to 1998, and 1998 to 2011.

| Land cover type (%) | Dinder | | | | Rahad | | | |
|---|---|---|---|---|---|---|---|---|
| | 1972 | 1986 | 1998 | 2011 | 1972 | 1986 | 1998 | 2011 |
| Bare area | 5 | 1 | 0 | 2 | 6 | 5 | 0 | 3 |
| Woodland | 42 | 23 | 27 | 14 | 35 | 14 | 21 | 14 |
| Shrubland | 23 | 43 | 21 | 36 | 30 | 32 | 13 | 15 |
| Grassland | 16 | 18 | 5 | 1 | 11 | 22 | 9 | 1 |
| Cropland | 14 | 15 | 45 | 47 | 18 | 26 | 55 | 68 |

### 4.1.1    Calibration and validation of the hydrological model results

To assess the reliability of the SBRE products, validation is done with the use of ground measurements at four gauges in which observed data are available. Two gauges (Gonder and Bahir Dar) are located nearby the upstream part of the catchments in the Ethiopian plateau, while the other two (Gedarif and Al-Hawata) are located at the most downstream part of the catchment in the Sudan low land. The validation is performed at annual time step. The results show that the difference of RFE against ground measurements has no consistent patterns. TRMM and CHIRPS have shown no consistent patterns at the lowland (Gedarif and Al-Hawata), but both products are consistent and overestimate rainfall at the Ethiopian highland (Gonder and Bahir Dar) in all years except 2007 (Figure 5). Since both the Dinder and the Rahad derive their main flow from the Ethiopian highlands, products with consistent patterns at highlands will be more suitable for running hydrologic models in this catchment. From these findings, one can conclude that the CHIRPS v2.0 and TRMM 3B42 v7 are more suitable than RFE 2.0 for running hydrologic model. Comparing CHIRPS v2.0 and TRMM 3B42 v7, it is clear that CHIRPS v2.0 has less overestimation of rainfall. Thus, CHIRPS v2.0 is the best product to be used as a forcing data for hydrologic model in the Dinder and Rahad basins.

The NSE and $R^2$ ranged from 0.4 to 0.80 and 0.50 to 0.80, respectively for the daily calibration and validation for the three precipitation products at Al-Gewisi station on the Dinder River and Al-Hawata station on the Rahad River (Fig. 6 and 7). At Al-Gewisi station, the large underestimation in the first validation period for CHIRPS can be attributed to the underestimation of rainfall by CHIRPS in 2007 at both Gonder and Bahir Dar (Figure 4). While at the same time CHIRPS overestimates rainfall in all years from 2001 to 2006. Therefore, calibration of the hydrologic model (during the period 2002-2005) resulted in underestimation of river flow in 2007. On the other hand, at Al-Hawata station, the difference between observed and model flow in the first period of validation (i.e.2008) is likely due to an error either in the input data or the observed flow values or a combination of both.

In general, the calibration results indicate that CHIRPS 2.0 is the best products over rugged terrain with complex rainfall patterns in the D&R basins. This result is in agreement with Hessels (2015), who compared and validated 10 open-access and spatially distributed satellite rainfall products over the Nile Basin and found that CHIRPS is the best product to be used in the Nile Basin. The modelling results show that the approach is reasonably good and therefore can be used in predicting runoff at a sub-basin level. Then the model was used to simulate the impact of LULCC on streamflow by running the model using land cover from different periods of time (1972, 1986, 1998 and 2011) and keeping precipitation (CHIRPS), evapotranspiration and other model parameters without change.

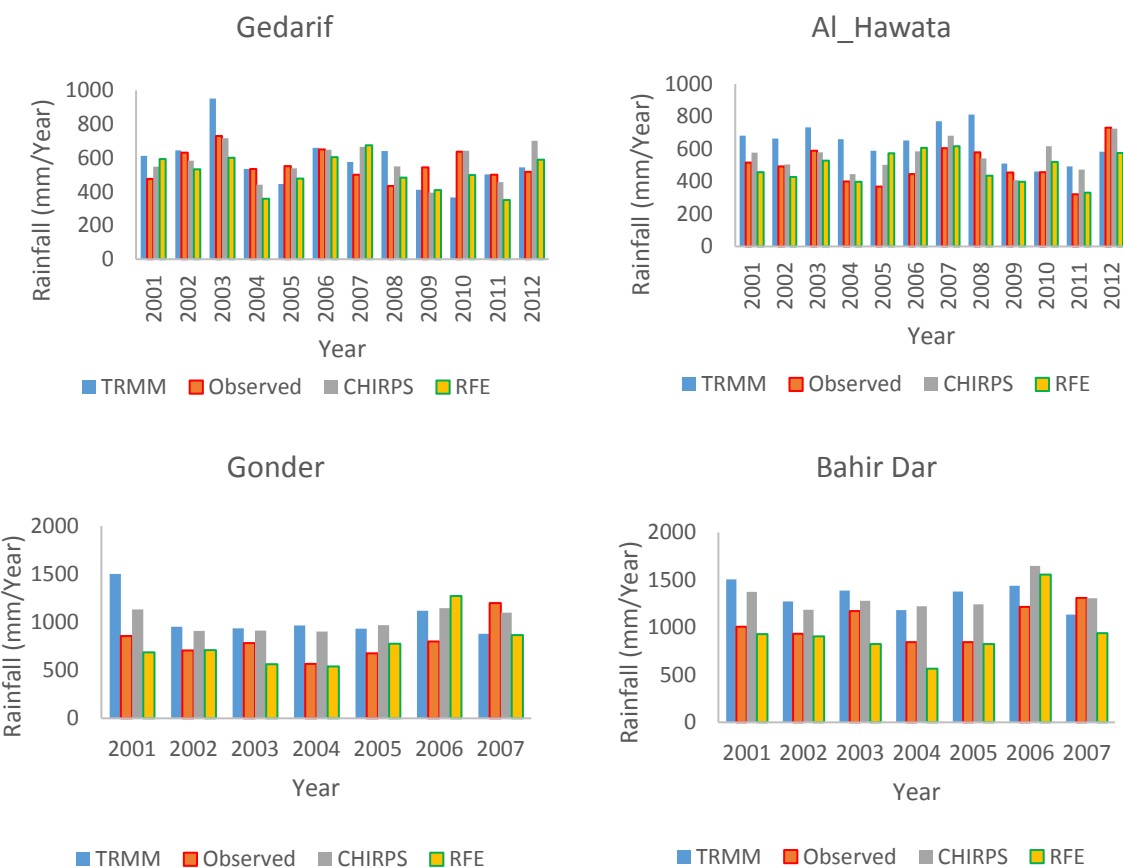

Figure 5: Comparison of SBRE products with ground measurements at four locations

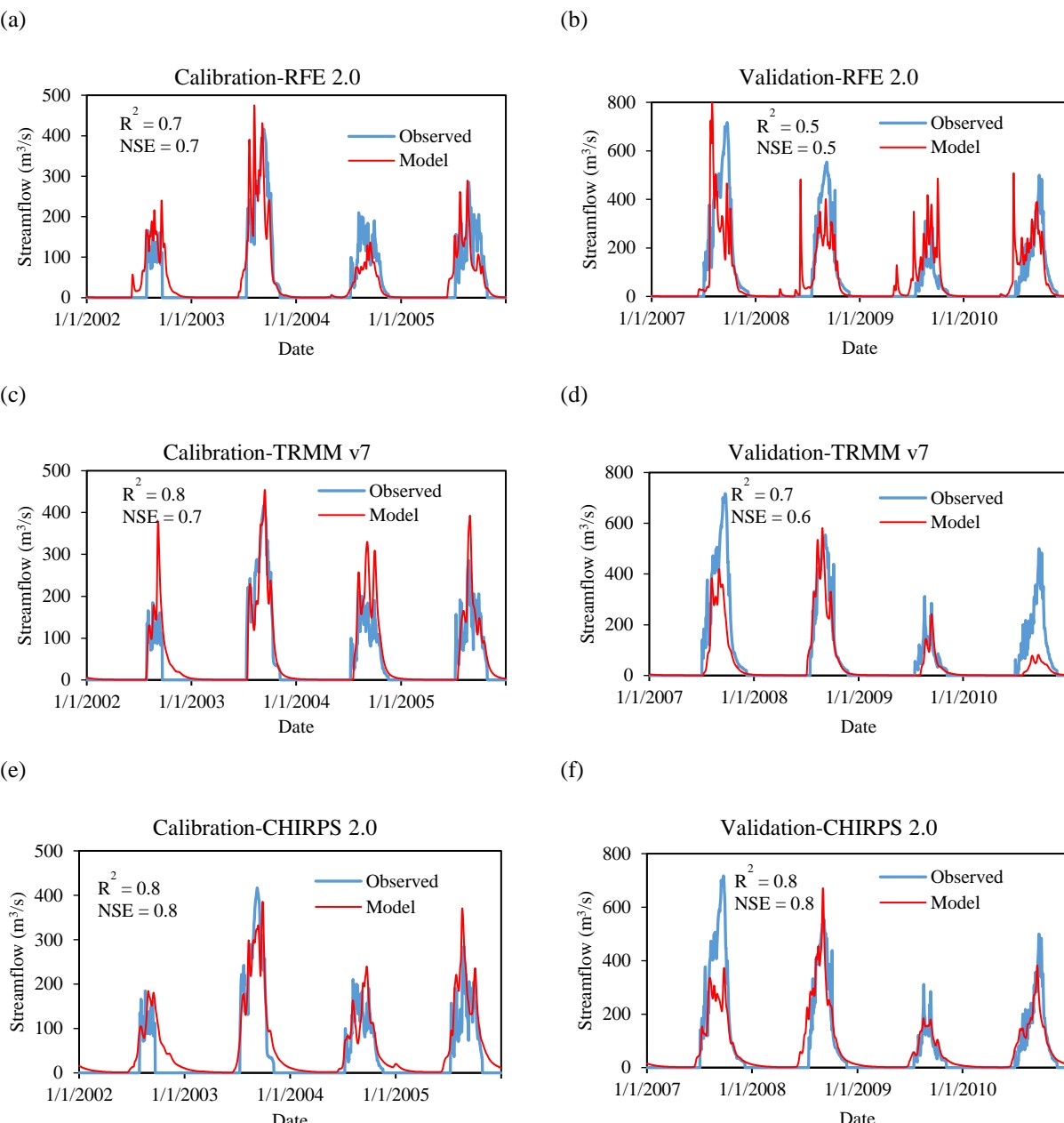

**Figure 6**. Calibration and validation results at Al-Gewisi station on Dinder River (a) and (b) for RFE, (c) and (d) for TRMM and (e) and (f) for CHIRPS.

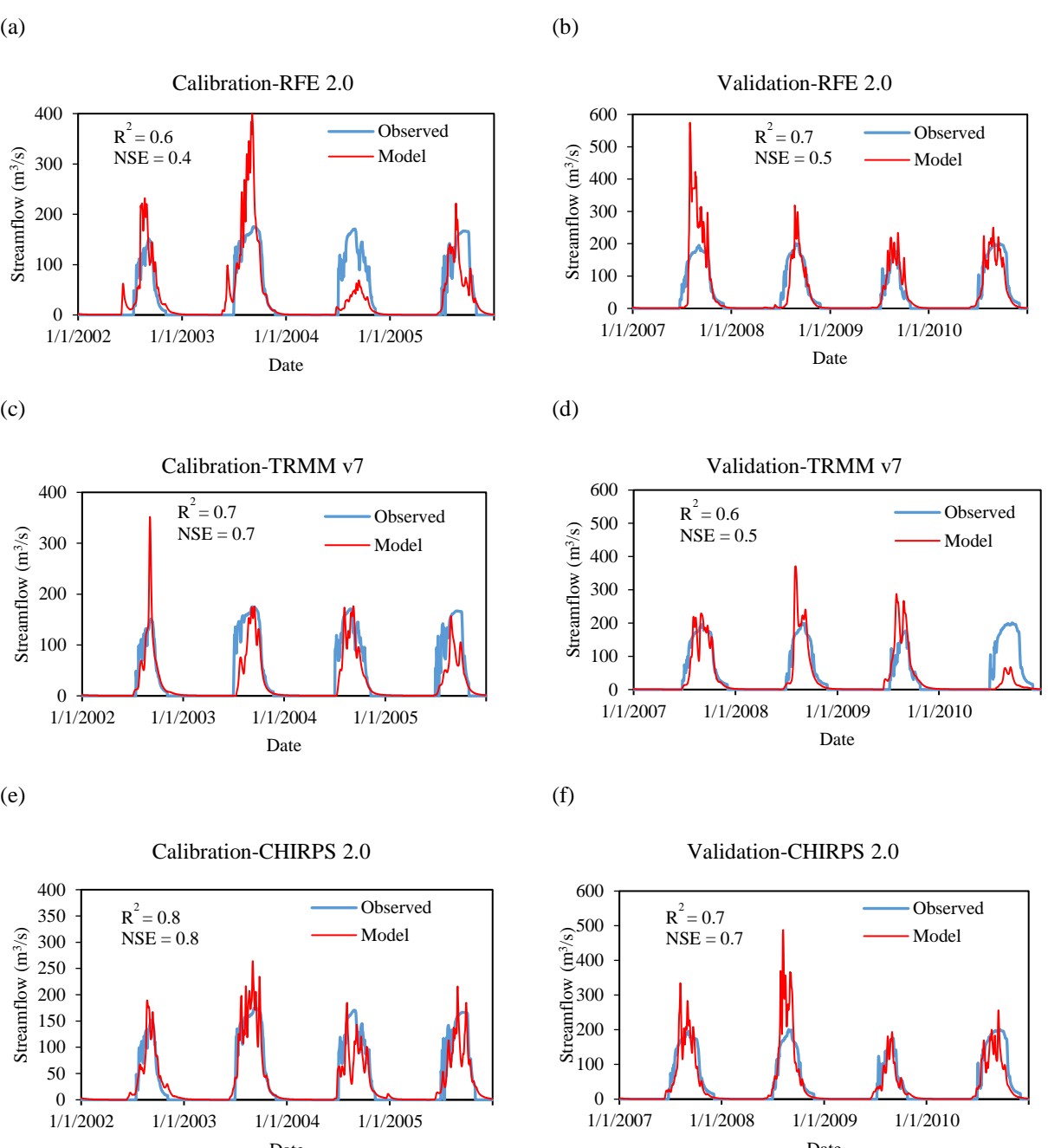

**Figure 7.** Calibration and validation results at Al-Hawata station on Rahad River (a) and (b) for RFE, (c) and (d) for TRMM and (e) and (f) for CHIRPS.

## 4.2    Streamflow response under land cover conversions

5    After the calibration and validation of the WFlow, the model has been run using different land use with fixed model parameters. First with land use from 1972; second with land use from 1986; third with land use from 1998; and fourth with land use from 2011. Then the output flows from the four land uses were compared. We note that the rainfall (CHIRPS) and PET for the period 2001-2012 were used with the 1972, 1986, 1998 and 2011 land uses to identify hydrological impacts of changes in land cover explicitly.

10    The WFlow result indicates that streamflow is affected by LULCC in both the Dinder and the Rahad Rivers. The effect of LULCC is much larger in the Rahad than in the Dinder.  In the Rahad basin, the simulated streamflow showed low peak flow with land use of 1972 and high flow with land use of 2011. Woodland and shrubland are dominants in 1972 and occupied 35%

and 30% of the upper catchment area respectively. While cropland is the dominant land cover type in 2011 which occupied 68%. Woodland and shrubland have high porosity and they delayed the release of water to the catchment outlet. Woodland removal implies less infiltration due to a decrease in soil permeability and less interception of rainfall by the tree canopies and thus more runoff and high flow peaks. The daily streamflow of the Dinder and the Rahad as results from different LULC are shown in Fig. 8. Figure 9.a shows the simulated streamflow of the Rahad River as a result from land covers of 1972, 1986, 1998 and 2011. Annual streamflow increased by 75% between 1972 and 1986, but is followed by a decrease of 45% between 1986 and 1998. The increase of streamflow could be a result of a decrease in woodland by 60% from 35% in 1972 to 14% in 1986 associated with an increase in cropland and grassland. Cropland has increased by 44% from 18% in 1972 to 26% in 1986 and grassland has increased by 100% from 11% in 1972 to 22% in 1986. This increase of grassland thus decreases water infiltration due to soil compaction caused by grazing which causes both higher runoff and an increase in annual streamflow magnitude. During the period 1986-1998, cropland and woodland showed a significant increase by 113% and 53%, respectively, while the remaining categories showed declines. During the period 1998-2011, the annual streamflow increased by 65% and corresponds with results on increases in the percentage of bare land, cropland, and shrubland by 754%, 23% and 15%, respectively, while a decrease in woodland and grassland by 37%, and 94%, respectively.

Similar to the Rahad, the simulated streamflow of the Dinder River showed low peak flow with land use of 1972 and relatively high flow with land use of 2011. Woodland is dominant in 1972 and occupied 42% of the total catchment area. While cropland is the dominant land cover type in 2011 which occupied 47%. Figure 9.b shows the simulated annual streamflow of the Dinder River as a result from land covers of 1972, 1986, 1998 and 2011. Annual streamflow increased by 20% between 1972 and 1986 but is followed by a decrease of 9% between 1986 and 1998. This could be a result of a decrease in woodland by 43% from 42% in 1972 to 23% in 1986 associated with an increase in shrubland, grassland and cropland by 83%, 10% and 6%, respectively. During the period 1986-1998, cropland and woodland increased by 192% and 16%, respectively, while the remaining categories showed declines. Over the period 1998-2011, the annual streamflow increased by 52% and corresponds with findings on increases in the percentage of bare land, cropland, and shrubland by 360%, 4% and 71%, respectively, while a decrease in woodland and grassland by 50%, and 76%, respectively. The decrease in percentage change of bare area over the period 1986-1998, beside the increase in woodland in both the Dinder and the Rahad basins indicate that the environment was recovering from the severe drought of 1984/1985.

In addition to the streamflow response to LULCC, evapotranspiration (ET) is another important component of the water balance that constitutes a major determinant of the amounts of water draining from different land cover type within the catchment. The ET result shows high rates of actual evapotranspiration (AET) when running the model with land cover of the years 1972 and 1998 at both the sub-catchments and the entire catchment (Table 4 and 5). This can be attributed to the large percentage coverage of woodland in 1972 and 1998 compared to land cover of 1986 and 2011 (please refer to table 3). The lowest AET is observed when running the model with land cover of 1986. This is likely due to the severe drought during the mid-1980s that limits the water availability and decreases the green coverage. Table 4 presents the changing in the annual average AET at sub-catchment level as a response to LULCC for the Dinder catchment. Table 5 shows the changes in water balance for the entire Dinder and Rahad catchments when running the hydrologic model with different LULC and fixed rainfall data for the periods (2001-2012).

Since both the Dinder and the Rahad rivers are seasonal, their flows are mainly depending on rainfall patterns and magnitudes. In addition to the effect of LULCC on the streamflow, Figure 10 shows that the annual variability of rainfall is another factor affecting the annual patterns of the streamflow.

(a)                                                    (b)

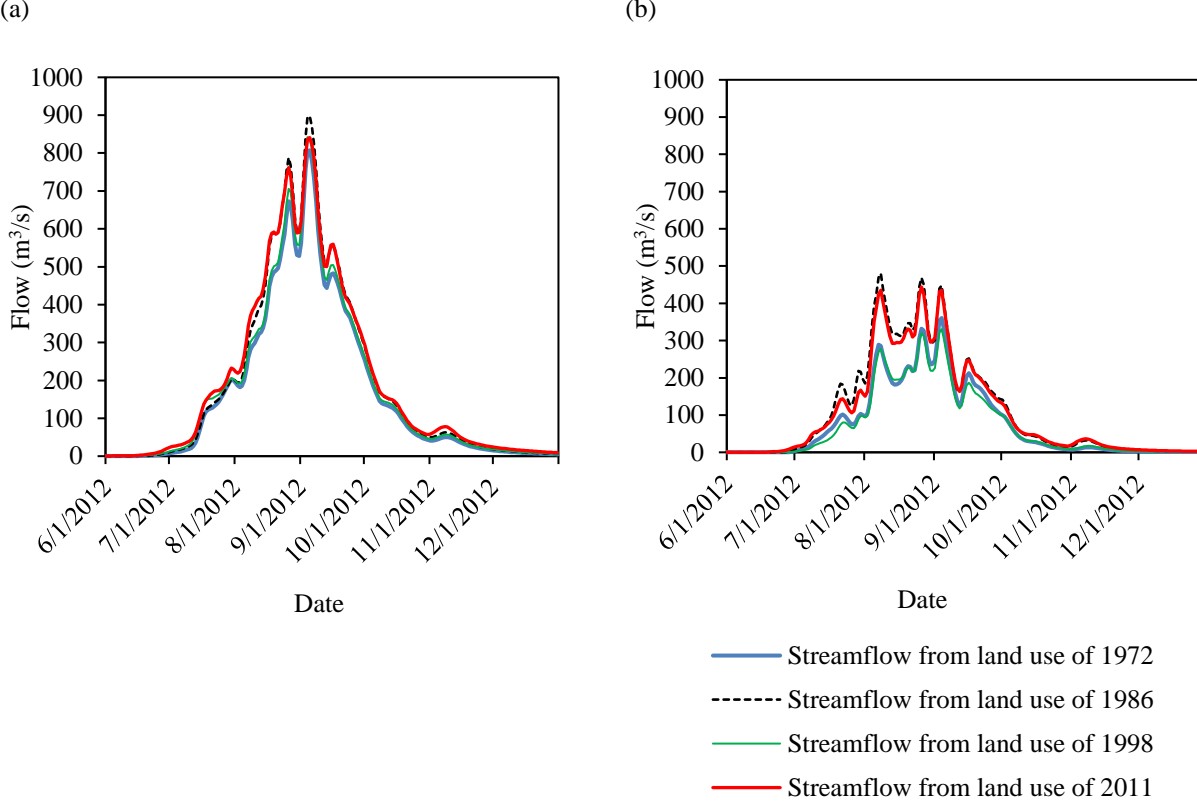

**Figure 8.** Daily streamflow results from the WFlow model at (a) Al-Gewisi station on the Dinder River and (b) Al-Hawata station on the Rahad River based on land use from 1972, 1986, 1998 and 2011 for the year 2012 as an example.

(a)

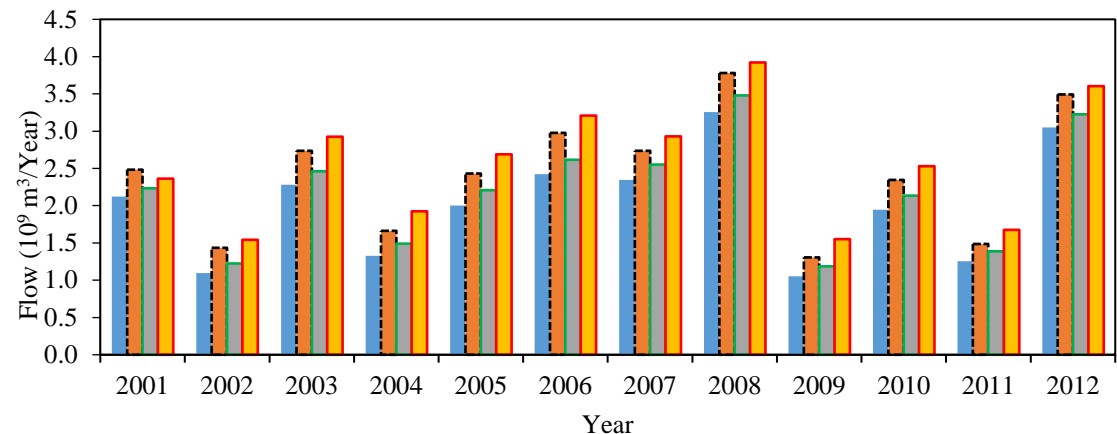

(b)

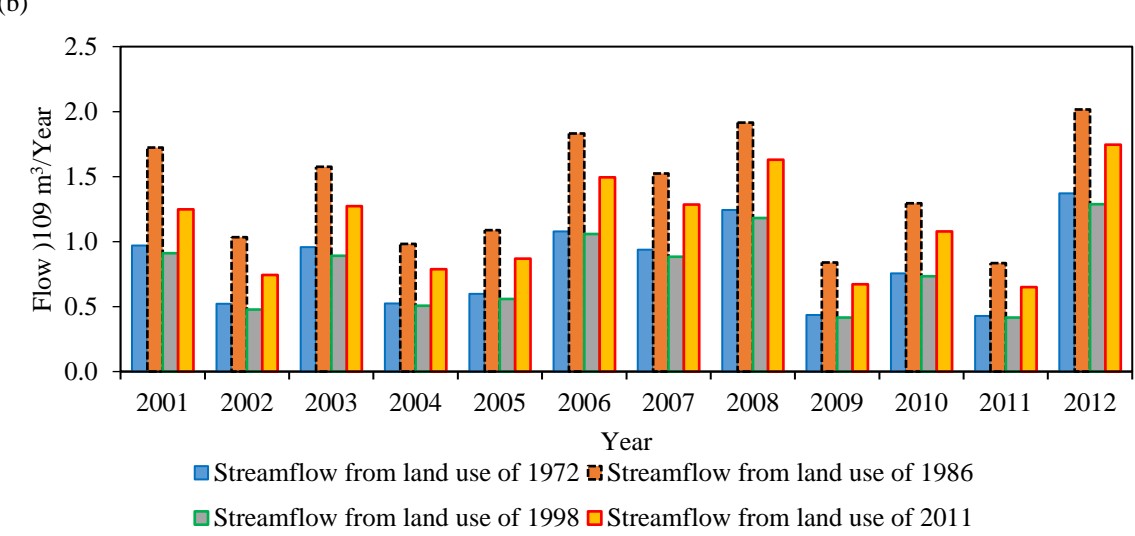

■ Streamflow from land use of 1972 ■ Streamflow from land use of 1986
■ Streamflow from land use of 1998 ■ Streamflow from land use of 2011

**Figure 9.** Annual streamflow results from the WFlow model at (a) Al-Gewisi station on the Dinder River and (b) Al-Hawata station on the Rahad River based on land use from 1972, 1986, 1998 and 2011

Table 4: Annual average AET as a response to LULCC at the sub-catchments level for the Dinder catchment (1972-1986).

| Year | AET from land cover of 1972 (mm) | | | | AET from land cover of 1986 (mm) | | | |
|---|---|---|---|---|---|---|---|---|
| | Al-Gewisi | Musa | Gelagu | Upper Dinder | Al-Gewisi | Musa | Gelagu | Upper Dinder |
| 2001 | 558 | 583 | 626 | 464 | 426 | 424 | 396 | 288 |
| 2002 | 443 | 456 | 535 | 510 | 322 | 317 | 306 | 312 |
| 2003 | 564 | 639 | 642 | 486 | 425 | 469 | 405 | 312 |
| 2004 | 455 | 502 | 573 | 500 | 326 | 354 | 340 | 311 |
| 2005 | 504 | 547 | 575 | 505 | 376 | 396 | 358 | 323 |
| 2006 | 527 | 576 | 632 | 545 | 396 | 414 | 406 | 359 |
| 2007 | 598 | 602 | 618 | 564 | 468 | 444 | 400 | 382 |
| 2008 | 593 | 689 | 703 | 576 | 459 | 513 | 471 | 392 |
| 2009 | 421 | 482 | 519 | 516 | 310 | 343 | 302 | 323 |
| 2010 | 536 | 566 | 606 | 520 | 412 | 415 | 383 | 331 |
| 2011 | 470 | 467 | 554 | 530 | 350 | 327 | 329 | 332 |
| 2012 | 636 | 679 | 684 | 542 | 500 | 504 | 450 | 353 |

Table 5:   Water balance of the Dinder and Rahad catchments applying different LULC

| Dinder catchment | | Land cover of 1972 | | Land cover of 1986 | | Land cover of 1998 | | Land cover of 2011 | |
|---|---|---|---|---|---|---|---|---|---|
| Year | Rainfall (mm) | AET (mm) | Streamflow (mm) | AET (mm) | Streamflow (mm) | AET (mm) | Streamflow (mm) | AET (mm) | Streamflow (mm) |
| 2001 | 816 | 558 | 258 | 383 | 433 | 432 | 384 | 496 | 320 |
| 2002 | 663 | 486 | 177 | 314 | 349 | 364 | 299 | 430 | 233 |
| 2003 | 847 | 583 | 264 | 403 | 444 | 449 | 397 | 519 | 327 |
| 2004 | 703 | 507 | 195 | 333 | 370 | 374 | 329 | 451 | 252 |
| 2005 | 768 | 532 | 236 | 363 | 405 | 414 | 354 | 479 | 289 |
| 2006 | 835 | 570 | 265 | 394 | 441 | 441 | 395 | 513 | 322 |
| 2007 | 876 | 595 | 280 | 424 | 452 | 476 | 400 | 540 | 336 |
| 2008 | 929 | 640 | 289 | 459 | 470 | 509 | 420 | 582 | 347 |
| 2009 | 659 | 484 | 175 | 319 | 340 | 363 | 297 | 435 | 225 |
| 2010 | 817 | 557 | 260 | 385 | 432 | 432 | 386 | 505 | 312 |
| 2011 | 710 | 505 | 205 | 334 | 376 | 377 | 333 | 454 | 256 |
| 2012 | 972 | 635 | 337 | 452 | 520 | 498 | 474 | 579 | 393 |
| Rahad catchment | | Land cover of 1972 | | Land cover of 1986 | | Land cover of 1998 | | Land cover of 2011 | |
| Year | Rainfall (mm) | AET (mm) | Streamflow (mm) | AET (mm) | Streamflow (mm) | AET (mm) | Streamflow (mm) | AET (mm) | Streamflow (mm) |
| 2001 | 724 | 409 | 315 | 290 | 434 | 398 | 326 | 309 | 416 |
| 2002 | 641 | 398 | 243 | 271 | 370 | 383 | 258 | 291 | 350 |
| 2003 | 755 | 450 | 305 | 323 | 432 | 434 | 322 | 342 | 413 |
| 2004 | 609 | 360 | 249 | 231 | 378 | 338 | 270 | 244 | 364 |
| 2005 | 656 | 399 | 258 | 267 | 389 | 378 | 278 | 285 | 372 |
| 2006 | 782 | 450 | 332 | 324 | 457 | 431 | 351 | 336 | 446 |
| 2007 | 774 | 473 | 301 | 344 | 430 | 456 | 319 | 363 | 411 |
| 2008 | 754 | 438 | 315 | 313 | 441 | 415 | 338 | 322 | 431 |
| 2009 | 581 | 352 | 229 | 220 | 361 | 333 | 248 | 238 | 343 |
| 2010 | 744 | 449 | 295 | 319 | 425 | 431 | 313 | 335 | 409 |
| 2011 | 610 | 369 | 241 | 235 | 375 | 348 | 262 | 252 | 358 |
| 2012 | 873 | 507 | 366 | 381 | 492 | 485 | 388 | 390 | 483 |

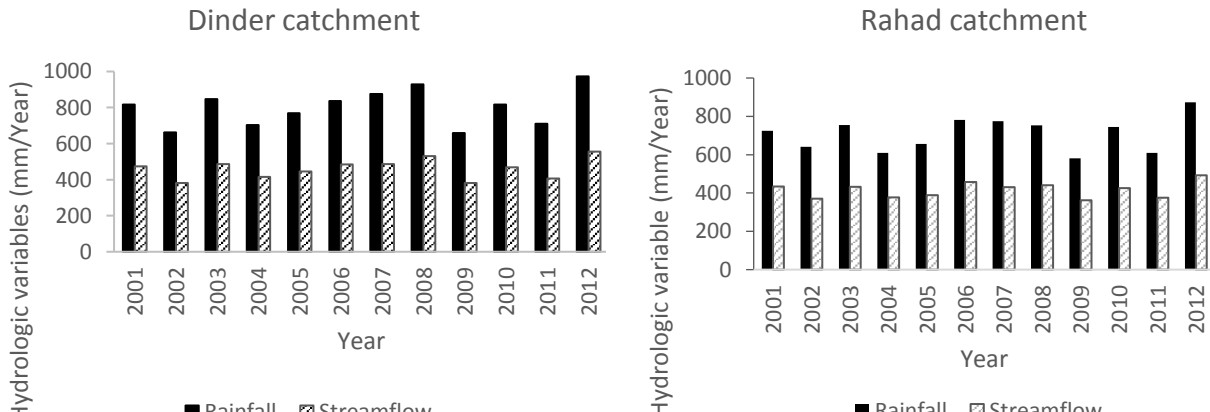

Figure 10: Annual average rainfall and streamflow patterns and magnitudes for the years (2001-2012).

10    **4.3    Streamflow analysis with IHA**

Since both Dinder and Rahad are seasonal rivers (July-November) and its floodplains including the mayas are mainly depending on floods, the streamflow analysis is focused on the flows during the months of high flows and the indicators describing the hydrological high extremes. The investigated streamflow variables are a subset of the 32 indicators proposed

by Richter et al. (1996) under the Range of Variability approach (RVA) that characterizes the natural flow regime of a river into five categories of magnitude, timing, duration, frequency and rate of change. In this section, we analyzed the modelled streamflow as a result from LULC of 1972, 1986, 1998 and 2011.

### 4.3.1 Magnitude of monthly flow

The general pattern of median monthly flow of the Rahad River (Fig. 11.a) at Al-Hawata station during 1972-1986 is that the median flow increased in all months of flow (July-November) with an average of 83% per month. In contrast, the median monthly flow decreased in all months during the period 1986-1998 with an average of 45% per month. Similar to the period from 1972-1986, the median monthly flow during 1998-2011 increased by an average of 65% per month.

In comparison to Rahad, the Dinder median monthly flow (Fig. 11.b) at Al-Gewisi station during 1972-1986 increased in all months of flow by an average of 21% per month. In contrast, the median monthly flow decreased in all months during the period 1986-1998 with an average of 6% per month. Likewise, to the period from 1972-1986, the median monthly flow during 1998-2011 increased by an average of 17% per month. Alterations of the monthly flow magnitude, particularly during the months of high flows (August-October) is likely affecting habitat availability on floodplains, which may lead to decrease and/or disappearance of native flora and increase in non-natives flora that might not be suitable for the herbivores wildlife that dwells in the DNP.

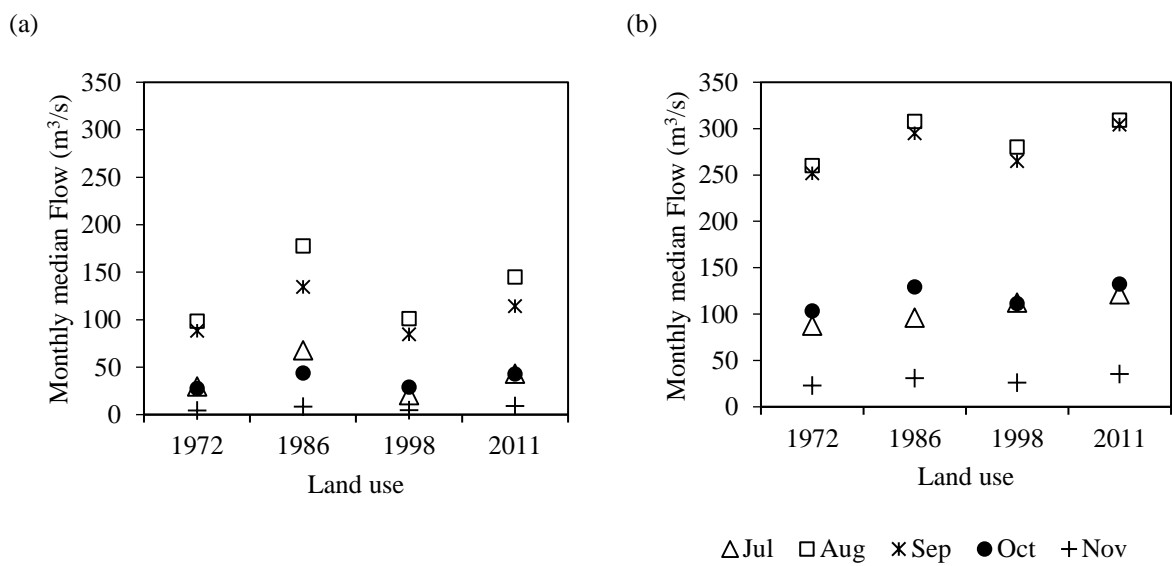

**Figure 11.** The monthly median flow (a) for Rahad River and (b) for Dinder River

### 4.3.2 Magnitude of river extreme floods

Extreme floods are important in re-forming both the biological and physical structure of a river and its associated floodplain. Extreme floods are also important in forming key habitats such as oxbow lakes and floodplain wetlands. The pattern of the extreme flow is vital for the filling of wetland mayas of the DNP. Therefore, annual flow maxima of 1, 7, 30 and 90-day have been investigated. The median maxima are presented in Fig. 12. In general, all results have shown that the maxima are significantly affected by LULCC. In Rahad median flow maxima for 1, 7, 30 and 90-day intervals from the land use of 1986 are 51%, 56%, 67%, and 68%, respectively higher than the maxima from the land use of 1972. Likewise, median flow maxima for 1, 7, 30 and 90-day intervals from the land use of 2011 are 32%, 33%, 36%, and 39%, respectively higher than the maxima

from the land use of 1998. In contrast, median flow maxima for 1, 7, 30 and 90-day intervals from the land use of 1998 are 39 %, 39%, 42%, and 42%, respectively lower than the maxima from the land use of 1986.

In the Dinder River the effect of LULCC on streamflow is not big as in Rahad River. This is likely due to the large expansion in cropland in the Rahad catchment to 68% of the total area compared to 47% in the Dinder catchment. The median flow maxima for 1, 7, 30 and 90-day intervals from the land use of 1986 are 19 %, 19%, 18%, and 18%, respectively higher than the maxima from the land use of 1972. Likewise, the median flow maxima for 1, 7, 30 and 90-day intervals from the land use of 2011 are 14 %, 13%, 14%, and 19% respectively higher than the maxima from the land use of 1998. In contrast, the median flow maxima for 1, 7, 30 and 90-day intervals from the land use of 1998 are 11 %, 11%, 10%, and 10%, respectively lower than the maxima from the land use of 1986. Peak flows are the critical aspects of the lateral connectivity between the Rahad and the Dinder rivers and its floodplains. Reduction of the magnitude of these high flow peaks during dry years (less than average) may reduce the ecological function of the mayas wetlands areas as breeding, nursery and feeding habitat for wildlife.

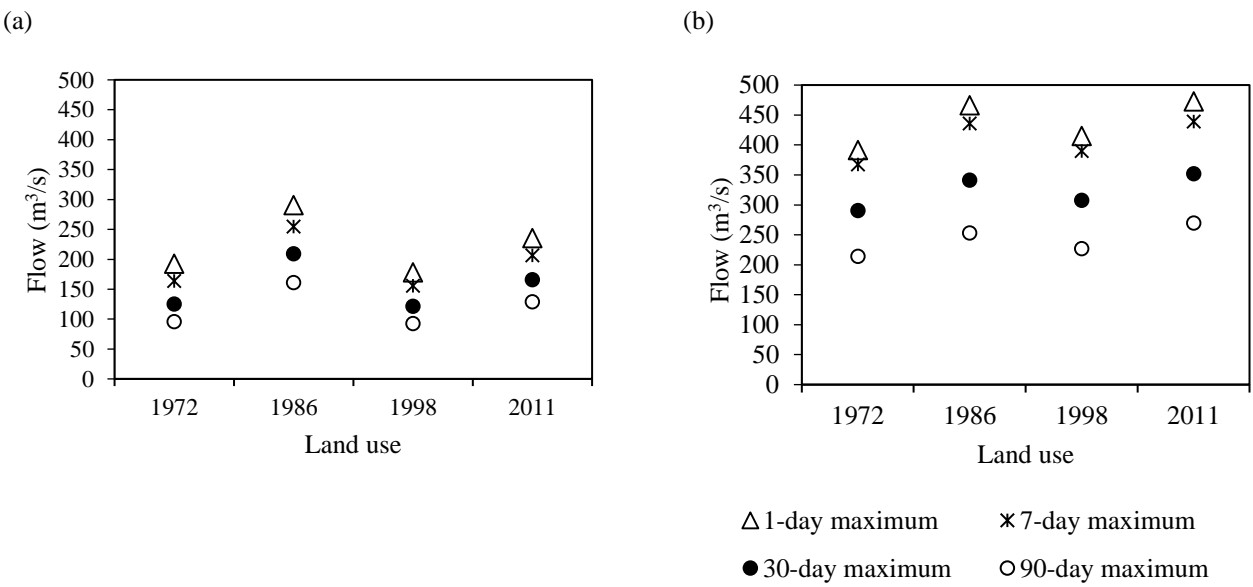

**Figure 12.** Median flow maxima for 1, 7, 30 and 90-day intervals from the land use of 1972, 1986, 1998 and 2011 for (a) Rahad River and (b) Dinder River.

### 4.3.3 Timing of annual extreme floods

Synchronization of annual flood with a variety of riverine and floodplain species life-cycle requirements is of likely high importance given the adaptation of species to their habitat. In the Rahad River, dates of the annual maxima as results from the land use of 1972, 1986, 1998 and 2011 occurred within the same three weeks (15 August – 02 September, Julian date (JD) 227–245). The annual maxima from the land use of 1986 is 18 days earlier than the annual maxima from land use of 1972. This could be attributed to land cover degradation and deforestation due to the devastating drought of 1984/1985 result in accelerating the runoff response.

In Dinder River, dates of the annual maxima are not affected by LULCC and occurred within the same two days (11 September – 12 September, Julian date (JD) 254–255).

#### 4.3.4 Rate of change in flow

The rate of change in flow can affect persistence and lifetime for both aquatic and riparian species (Poff et al. 1997), particularly in arid areas where streamflow usually changes rapidly in a very short time. Figure 13 shows the rate of flow-rises and flow-falls for both Rahad and Dinder. The median rate of flow-rises (positive differences between consecutive daily values) in Rahad River has increased by 74% from 2.73 (m³/s) /day in 1972 to 4.73 (m³/s) /day in 1986. In 1998 the median rate of flow-rises decreased by 50%, while increasing by 37% in 2011. Similarly, the median rate of flow-falls (negative differences between consecutive daily values) has increased by 88% from 0.12 (m³/s) /day in 1972 to 0.23 (m³/s) /day in 1986. In 1998 the median rate of flow-falls decreased by 37%, while increasing by 22% in 2011. Likewise, the median rate of flow-rises and flow-falls in the Dinder River follow the same pattern of the Rahad flow, but no significant changes were observed. This result shows that the fluctuation in rate of change in streamflow is strongly linked to LULCC, especially when analyzing the streamflow as a result from land use after a period of drought (e.g. land use of 1986).

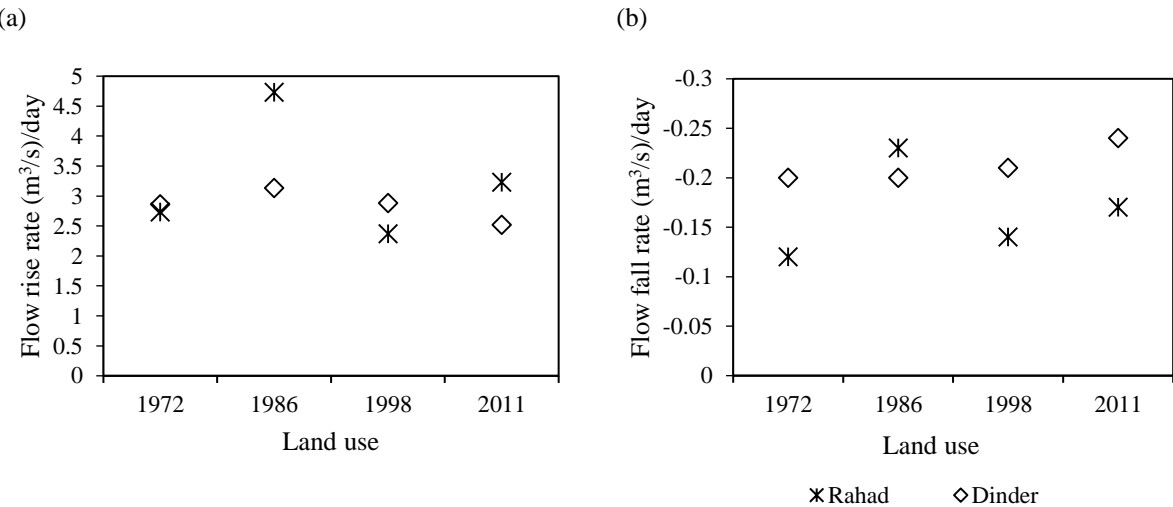

**Figure 13.** The rate of flow rises (a) and falls (b) as a response to land use of 1972, 1986, 1998 and 2011 for both Rahad and Dinder Rivers (negative sign in the vertical axis indicates downward direction of flow).

### 5    Conclusion

For assessing the changes in land cover, four remote sensing images were used for the years 1972, 1986, 1998 and 2011. The accuracy assessment with supervised land cover classification shows that the classification results are reliable. The land cover changes in the D&R are assessed by image comparison and the results showed that the dominant process is the relatively large decrease of woodland and the large increase of cropland. Results of LULCC detection between 1972 and 2011 indicate a significant decrease of woodland and an increase of cropland. Woodland decreased from 42% to 14% and from 35% to 14% for Dinder and Rahad respectively. Cropland increased from 14% to 47% and from 18% to 68% in Dinder and Rahad respectively. The rate of deforestation is high during the period 1972-1986 and probably is due to the severe drought during 1984/1985, expansion in agricultural activities and increased demand for wood for fuel, construction and other human needs due to the increase in population. On the other hand, increasing in woodland during the period between 1986 and 1998 is probably due to reforestation activities in the basin. Nevertheless, the magnitude of deforestation is still much larger than the reforestation. The cropland expansion over the period 1986 to 1998 is larger than the expansion over the period 1998 to 2011, suggests that most of the areas that are suitable for cultivation have most likely been occupied, or the land tenure regulations have controlled the expansion of cultivation by local communities.

The results of the hydrological model indicate that streamflow is affected by LULCC in both the Dinder and the Rahad Rivers. The effect of LULCC on streamflow is significant during 1986 and 2011 particularly in the Rahad River. This could be attributed to the severe drought during 1984/1985 and the large expansion in cropland in the Rahad catchment to 68% of the total area.

The IHA analysis indicated that the flow of the Dinder and the Rahad Rivers was associated with significant upward and downward alterations in magnitude, timing and rate of change of river flows, as a result of LULCC. These alterations in the streamflow characteristics are likely to have significant effects on a range of species that depend on the seasonal patterns of flow. Therefore, alterations in the magnitude of the annual floods that decrease the water flowing to the mayas may reduce the production of native river-floodplain fauna and flora, and migration of animals that may be connected to mayas inundation.

***Competing interests*:** The third author is a member of the editorial board of the journal.

***Acknowledgements:*** This study was carried out as part of a Ph.D. research program of the first author entitled 'The impacts of land degradation on the Dinder and Rahad hydrology and morphology, and linkage to the ecohydrological system of the Dinder National Park, Sudan'', which is funded by the Netherland Fellowship Program (NFP). We also thank the Hydraulics Research Center of the Ministry of Water Resources and Electricity-Sudan for providing the hydrological data.

Appendix A:

*The wflow_sbm interception model*:

The analytical model of rainfall interception is based on Rutter's numerical model (see Gash, 1979; Gash et al., 1995, for a full description). The simplifications that Gash (1979) introduced allow the model to be applied on a daily basis. The amount of water needed to completely saturate the canopy ($P'$) is defined as:

$$P' = \frac{-\bar{R}S}{\bar{E}_w} ln \left[1 - \frac{\bar{E}_w}{\bar{R}}(1 - p - p_t)^{-1}\right] \qquad (A1)$$

where:

$\bar{R}$ = average precipitation on a saturated canopy [mm day$^{-1}$]

$\bar{E}_w$ = average evaporation from the wet canopy [mm day$^{-1}$]

$S$ = canopy storage capacity [mm]

$p$ = free throughfall coefficient: the proportion of rain which falls to the ground without sticking the canopy [-]

$p_t$ = proportion of rain that is diverted to stemflow [-]

Interception losses from the stems are calculated for days with $P \geq S_t/P_t$. $S_t$ (trunk water capacity [mm]) and $P_t$ are small and neglected in the wflow_sbm model. In applying the analytical model, saturated conditions are assumed to occur when the hourly rainfall exceeds a certain threshold. Often a threshold of 0.5 mm hr$^{-1}$ is used (Gash, 1979). $\bar{R}$ is calculated for all hours when the rainfall exceeds the threshold to give an estimate of the mean rainfall rate onto a saturated canopy. $E_w$ is then calculated using the Rutter model.

*The wflow_sbm soil water accounting scheme:*

Within the soil model, the soil is considered as a bucket with a certain depth ($Z_t$), divided into a saturated store (S) and an unsaturated store (U), the capacity of each is expressed in units of depth. The top of the saturated store forms a pseudo-water table at depth ($Z_i$) such that the value of (S) at any time is given by:

$$S = (z_t - z_i)(\theta_s - \theta_r) \qquad \text{(A2)}$$

Where:

$\theta_s$ and $\theta_r$ are the saturated and residual soil water contents, respectively.

The unsaturated store (U) is subdivided into storage ($U_s$) and deficit ($U_d$) which are also expressed in units of depth:

$$U_d = (\theta_s - \theta_r)z_i - U \qquad \text{(A3)}$$

and

$$U_s = U - U_d \qquad \text{(A4)}$$

The saturation deficit ($S_d$) for the whole soil profile is defined as:

$$S_d = (\theta_s - \theta_r)z_t - S \qquad \text{(A5)}$$

Infiltrating rainfall enters the unsturated store first. The transfer of water from the unsaturated store to the saturated store (st) is controlled by the saturated hydraulic conductivity $K_{sat}$ at depth ($Z_i$) and the ratio between $U_s$ and $S_d$.

$$st = K_{sat}\frac{U_s}{S_d} \qquad \text{(A6)}$$

As the saturation deficit becomes smaller, the rate of the transfer between the unsaturated and saturated stores increases. Saturated conductivity ($K_{sat}$) declines with soil depth (z) in the model according to:

$$K_{sat} = K_0\, e^{(-fz)} \qquad \text{(A7)}$$

where:
$K_0$ is the saturated conductivity at the soil surface [m day$^{-1}$] and;
$f$ is a scaling parameter [m$^{-1}$]
The scaling parameter f is defined by:

$$f = \frac{\theta_s - \theta_r}{M} \qquad \text{(A8)}$$

M is a soil parameter determining the decrease of saturated conductivity with depth [m].

The saturated store can be drained laterally via subsurface flow according to:

$$sf = K_0 tan(\beta) \, e^{-S_d/M} \qquad \text{(A9)}$$

where:

$\beta$ is element slope angle [deg.]

$sf$ is the calculated subsurface flow [$m^2 \, day^{-1}$]

The original SBM model does not include transpiration or a notion of capillary rise. In wflow_sbm transpiration is first taken from the saturated store if the roots reach the water table ($Z_i$). If the saturated store cannot satisfy the demand the unsaturated store is used next. First the number of wet roots (WR) is determined (going from 1 to 0) using a sigmoid function as follows:

$$WR = 1.0/(1.0 + e^{-SN(WT-RT)}) \quad \text{(A10)}$$

where:

SN is sharpness parameters

WT is water table [mm]

RT is rooting depth [mm]

The sharpness parameter (by default a large negative value, -80000) is a parameter determines if there is a stepwise output or a more gradual output (default is stepwise). Water Table is the level of the Water table in the grid cell below the surface and rooting depth is the maximum depth of the roots below the surface. For all values of water tables smaller that rooting depth a value of 1 is returned, if they are equal to rooting depth a value of 0.5 is returned, and if the water table is larger than the rooting depth a value of zero is returned. The returned wet roots (WR) fraction is multiplied by the potential evaporation (and limited by the available water in saturated store) to get the transpiration from the saturated part of the soil. Next the remaining potential evaporation is used to extract water from the unsaturated store.

Capillary rise is determined using the following approach: first the $K_{sat}$ is determined at the water table ($Z_i$); next a potential capillary rise is determined from the minimum of the $K_{sat}$, the actual transpiration taken from the unsaturated store, the available water in the saturated store and the deficit of the unsaturated store. Finally, the potential rise is scaled using the distance between the roots and the water table using:

$$CS = CSF/(CSF + z_i - RT) \quad \text{(A11)}$$

in which CS is the scaling factor to multiply the potential rise with, CSF is a model parameter (default = 100) and RT is the rooting depth. If the roots reach the water table (RT > Zi) CS is set to zero and thus setting the capillary rise to zero. A detailed description of the TOPOG_SBM model has been provided by R. A. Vertessy and H. Elsenbeer [1]."

Appendix B:

5    Wflow model parameter's description

| Parameter name in Wflow | Description | Unit |
|---|---|---|
| CanopyGapFraction | Gash interception model parameter: the free throughfall coefficient. Fraction of precipitation that does not hit the canopy directly | [-] |
| EoverR (E/R) | Gash interception model parameter. Ratio of average wet canopy evaporation rate over average precipitation rate. | [-] |
| MaxCanopyStorage | Canopy storage. Used in the Gash interception model | [mm] |
| FirstZoneCapacity | Maximum capacity of the saturated store. | [mm] |
| FirstZoneKsatVer | Saturated conductivity of the store at the surface. The M parameter determines how this decreases with depth. | [mm] |
| FirstZoneMinCapacity | Minimum capacity of the saturated store [mm] | [mm] |
| InfiltCapPath | Infiltration capacity of the compacted soil fraction of each gridcell. | [mm/day] |
| InfiltCapSoil | Infiltration capacity of the non-compacted soil fraction of each grid cell | [mm/day] |
| M | Soil parameter determining the decrease of saturated conductivity with depth. | [m] |
| N | Manning N parameter for the Kinematic wave function. | |
| N_river | Manning's parameter for cells marked as river | |
| LeafAreaIndex | Total one-side green leaf area per ground surface area. | [-] |
| Albedo | Reflectivity of earth surface: the ratio of radiation reflected to the radiation incident on a surface. | [-] |
| Beta | element slope angle | [degree] |
| rootdistpar | Sharpness parameter determine how roots are linked to water table. | [mm] |
| PathFrac | Fraction of compacted area per grid cell. | [-] |
| RootingDepth | Rooting depth of the vegetation. | [mm] |
| CapScale | Scaling factor in the Capilary rise calculations | [mm/day] |
| RunoffGeneratingGWPerc | Fraction of the soil depth that contributes to sub-cell runoff | [-] |
| thetaR | Residual water content. | [-] |
| thetaS | Water content at saturation (porosity). | [-] |

35

Appendix C:

Wflow model parameters calibrated values

**Albedo**

| Land cover | Sub-catchment | Soil type | Value |
|---|---|---|---|
| 1 | [0,> | [0,> | 0.40 |
| 2 | [0,> | [0,> | 0.20 |
| 3 | [0,> | [0,> | 0.16 |
| 4 | [0,> | [0,> | 0.26 |
| 5 | [0,> | [0,> | 0.25 |
| 6 | [0,> | [0,> | 0.10 |

**CanopyGapFraction**

| Land cover | Sub-catchment | Soil type | Value |
|---|---|---|---|
| 1 | [0,> | [0,> | 1.0 |
| 2 | [0,> | [0,> | 0.2 |
| 3 | [0,> | [0,> | 0.6 |
| 4 | [0,> | [0,> | 0.5 |
| 5 | [0,> | [0,> | 0.4 |
| 6 | [0,> | [0,> | 0.5 |

**EoverR**

| Land cover | Sub-catchment | Soil tpye | Value |
|---|---|---|---|
| 1 | [0,> | [0,> | 0.0 |
| 2 | [0,> | [0,> | 0.3 |
| 3 | [0,> | [0,> | 0.2 |
| 4 | [0,> | [0,> | 0.2 |
| 5 | [0,> | [0,> | 0.1 |
| 6 | [0,> | [0,> | 0.0 |

**FirstZoneCapacity**

| Land cover | Sub-catchment | Soil type | Value |
|---|---|---|---|
| [0,> | [0,> | 1 | 44500 |
| [0,> | [0,> | 2 | 42000 |
| [0,> | [0,> | 3 | 44500 |
| [0,> | [0,> | 4 | 39000 |
| [0,> | [0,> | 5 | 44000 |
| [0,> | [0,> | 6 | 42000 |
| [0,> | [0,> | 7 | 44500 |

**FirstZoneKsatVer**

| Land cover | Sub-catchment | Soil type | Value |
|---|---|---|---|
| [0,> | [0,> | 1 | 511 |
| [0,> | [0,> | 2 | 600 |
| [0,> | [0,> | 3 | 543 |
| [0,> | [0,> | 4 | 525 |
| [0,> | [0,> | 5 | 586 |
| [0,> | [0,> | 6 | 576 |
| [0,> | [0,> | 7 | 540 |

**FirstZoneMinCapacity**

| Land cover | Sub-catchment | Soil type | Value |
|---|---|---|---|
| [0,> | [0,> | 1 | 125 |
| [0,> | [0,> | 2 | 50 |
| [0,> | [0,> | 3 | 137.5 |
| [0,> | [0,> | 4 | 33 |
| [0,> | [0,> | 5 | 87.5 |
| [0,> | [0,> | 6 | 60 |
| [0,> | [0,> | 7 | 70 |

**InfiltCapPath**

| Land cover | Sub-catchment | Soil type | Value |
|---|---|---|---|
| [0,> | [0,> | 1 | 5 |
| [0,> | [0,> | 2 | 21 |
| [0,> | [0,> | 3 | 5 |
| [0,> | [0,> | 4 | 32 |
| [0,> | [0,> | 5 | 34 |
| [0,> | [0,> | 6 | 5 |
| [0,> | [0,> | 7 | 21 |

**InfiltCapSoil**

| Land cover | Sub-catchment | Soil type | Value |
|---|---|---|---|
| [0,> | [0,> | 1 | 24 |
| [0,> | [0,> | 2 | 103 |
| [0,> | [0,> | 3 | 24 |
| [0,> | [0,> | 4 | 158 |
| [0,> | [0,> | 5 | 170 |
| [0,> | [0,> | 6 | 100 |
| [0,> | [0,> | 7 | 103 |

**LeafAreaIndex**

| Land cover | Sub-catchment | Soil type | Value |
|---|---|---|---|
| 1 | [0,> | [0,> | 0.0 |
| 2 | [0,> | [0,> | 8.8 |
| 3 | [0,> | [0,> | 7.0 |
| 4 | [0,> | [0,> | 0.6 |
| 5 | [0,> | [0,> | 0.7 |
| 6 | [0,> | [0,> | 0.0 |

**M**

| Land cover | Sub-catchment | Soil type | Value |
|---|---|---|---|
| [0,> | [0,> | 1 | 100 |
| [0,> | [0,> | 2 | 87 |
| [0,> | [0,> | 3 | 100 |
| [0,> | [0,> | 4 | 77 |
| [0,> | [0,> | 5 | 100 |
| [0,> | [0,> | 6 | 100 |
| [0,> | [0,> | 7 | 100 |

**MaxCanopyStorage**

| Land cover | Sub-catchment | Soil type | Value |
|---|---|---|---|
| 1 | [0,> | [0,> | 0.00 |
| 2 | [0,> | [0,> | 0.336 |
| 3 | [0,> | [0,> | 0.21 |
| 4 | [0,> | [0,> | 0.25 |
| 5 | [0,> | [0,> | 0.34 |
| 6 | [0,> | [0,> | 0.00 |

**N**

| Land cover | Sub-catchment | Soil type | Value |
|---|---|---|---|
| 1 | [0,> | [0,> | 0.42 |
| 2 | [0,> | [0,> | 0.80 |
| 3 | [0,> | [0,> | 0.70 |
| 4 | [0,> | [0,> | 0.65 |
| 5 | [0,> | [0,> | 0.80 |
| 6 | [0,> | [0,> | 0.12 |

**PathFrac**

| Land cover | Sub-catchment | Soil type | Value |
|---|---|---|---|
| [0,> | [0,> | 1 | 0.06 |
| [0,> | [0,> | 2 | 0.09 |
| [0,> | [0,> | 3 | 0.05 |
| [0,> | [0,> | 4 | 0.06 |
| [0,> | [0,> | 5 | 0.06 |
| [0,> | [0,> | 6 | 0.07 |
| [0,> | [0,> | 7 | 0.08 |

**RootingDepth**

| Land cover | Sub-catchment | Soil type | Value |
|---|---|---|---|
| 1 | [0,> | [0,> | 1000 |
| 2 | [0,> | [0,> | 1800 |
| 3 | [0,> | [0,> | 1400 |
| 4 | [0,> | [0,> | 1600 |
| 5 | [0,> | [0,> | 200 |
| 6 | [0,> | [0,> | 0 |

**thetaR**

| Land cover | Sub-catchment | Soil type | Value |
|---|---|---|---|
| [0,> | [0,> | 1 | 0.15 |
| [0,> | [0,> | 2 | 0.09 |
| [0,> | [0,> | 3 | 0.19 |
| [0,> | [0,> | 4 | 0.09 |
| [0,> | [0,> | 5 | 0.11 |
| [0,> | [0,> | 6 | 0.09 |
| [0,> | [0,> | 7 | 0.08 |

**thetaS**

| Land cover | Sub-catchment | Soil type | Value |
|---|---|---|---|
| [0,> | [0,> | 1 | 0.5 |
| [0,> | [0,> | 2 | 0.2 |
| [0,> | [0,> | 3 | 0.5 |
| [0,> | [0,> | 4 | 0.3 |
| [0,> | [0,> | 5 | 0.4 |
| [0,> | [0,> | 6 | 0.2 |
| [0,> | [0,> | 7 | 0.2 |

**RunoffGeneratingGWPerc**

| Land cover | Sub-catchment | Soil type | Value |
|---|---|---|---|
| [0,> | [0,> | [0,> | 0.1 |

**rootdistpar**

| Land cover | Sub-catchment | Soil type | Value |
|---|---|---|---|
| [0,> | [0,> | [0,> | -80000 |

**N_River**

| Land cover | Sub-catchment | Soil type | Value |
|---|---|---|---|
| [0,> | [0,> | [0,> | 0.035 |

**Beta**

| Land cover | Sub-catchment | Soil type | Value |
|---|---|---|---|
| [0,> | [0,> | [0,> | 0.6 |

**CapScale**

| Land cover | Sub-catchment | Soil type | Value |
|---|---|---|---|
| [0,> | [0,> | [0,> | 100 |

Land cover: 1= Bare land, 2= woodland, 3= shrubland, 4= grassland, 5= cropland, 6= water bodies.
Soil type: 1= Vertisols, 2= Luvisols, 3= Nitisols, 4= Leptosols, 5= cambisols, 6= Alisols, 7= Fluvisols.

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
