# Peer review of "Analysis of streamflow response to land use land cover changes using satellite data and hydrological modelling: case study of Dinder and Rahad tributaries of the Blue Nile (Ethiopia/Sudan)"

_Hydrology and Earth System Sciences, 2017_

## Referee Comment (RC1) · Anonymous Referee #1 · 1 May 2017

Studies on the impacts of land use and land cover on streamflow are important, especially in developing countries and in Africa where the influence of human activities on water have been rampant and are increasing by the day. Studies have shown that increase in population and subsequent increase in anthropogenic activities have caused changes in streamflow. Availability of long term records of flow, and weather and climate (rainfall, minimum and maximum temperature and other parameters) is a problem in many developing countries. The use of satellite data as derived from different sensors and hydrological models provide information to water resource managers and decision makers on the status and approaches in resource management. This study

has attempted to address some of the aspects.

"The rainfall runoff process over the upper Dinder and Rahad basins (D&R) is complex, non-linear, and exhibits temporal and spatial variability". This needs a citation to support it. "However, the impact is often not well understood with locally obtained data such as observed flow". → The main issue here is the fact that data from local stations, is often not long enough or have periods of missing gaps. As it reads, it may look like the information from local stations is not understood.

citation in line should follow the author–date.

Please also note the supplement to this comment:
http://www.hydrol-earth-syst-sci-discuss.net/hess-2017-128/hess-2017-128-RC1-supplement.pdf

**Supplement:**

[revised manuscript text omitted]

---

## Referee Comment (RC2) · Anonymous Referee #2 · 29 May 2017

The paper shows the impact on streamflow due to land use and land cover changes in two tributaries of the Blue Nile River Basin. The tributaries - Dinder and Rahad - lie in Sudan where the hydrological data situation is sparse.

Therefore, satellite data are used for estimating precipitation and evapotranspiration. After calibration of the model at two discharge gauges, the hydrological model is applied for analyzing the impact of different land use changes on some streamflow indices. The topic is interesting and scientifically challenging.

However, the paper needs improvement before getting published. I will not correct

some English grammar / expressions. A final proofreading from a native speaker is still recommended.

The comments are split into two parts:

1. Remote sensing:

Page 5 Line 9: The exact dates for these four years are needed. This is very important information for a land cover change detection analysis, also a brief information (e.g. coverage of cloud) about the quality of selected Landsat data is also necessary.

Page 5 Line 29. This description is not right. TMPA is just a product of TRMM. There are many TRMM products, here please specify which one you used. I think it should be TRMM 3B42V7.

Page 5 Line 34: here it was described as CHIRPS available from 1981, but in the Table 1 it was mistakenly written from 1983. Additionally, CHRIPS provides daily data for the globe, please correct "Pentads" in Table 1 accordingly.

Page 5 Line 36: The "TRMM" should be corrected to "CHIRPS"

Table 1: These products have different spatial resolutions, the authors should explain how they processed such data (how to deal with the difference in spatial resolution) and used them as input to the model.

Figure 3: In the caption, "19986" should be "1986". The legend is quite abnormal, in the remote sensing analysis, crop is more commonly assigned to yellow color, while natural vegetation to green. I advise to change legend. The four land cover maps in Figure 3 shows quite remarked differences, and it seems no regular pattern, which needs more discussion and analysis about the quality of classified map. Normally there should be a pattern, because human activities follow rivers to convert natural vegetation to crop lands. I advise to use one or more matchup Google Earth High Resolution Images to further prove/evaluate the reliability of classified map.
2. Hydrological Modelling:

Figure 1. I recommend to insert the Blue Nile River in the upper right map.

Page 5, line 15: Why did you reclassify the 44 soil mapping units into 8 dominant soil groups? Was this necessary for the hydrological modelling?

Figure 2: The WFlow\_sbm model needs more explanation. How is runoff generation modelled? How is ETA calculated? Is there no interflow component?

Chapter 3.4: The IHA approach should be explained in more detail (add app. half page)

Table 2: Please explain the accuracy assessment. What means "producer" and "user"? Did you perform a cross validation analysis? The accuracy seems very high with little uncertainty for all classes. Can you prove this?

Chapter 4.1.1 / Figure 4 and Figure 5: A critical discussion about the calibration and its uncertainties is totally missing. Couldn't you assess the reliability of the RS data by ground truth measurements (rain gauges)? Please comment on that. Concerning the figures, there are great differences in the peak flows with reverse biases. For instance, at Al-Gewisi station, you get a large underestimation in the first validation period for CHIRPS, whereas you get a large underestimation for the same time period and RS method for the Al-Hawata station. There are many contrary results comparing the two figures. Please discuss this issue. Moreover, did you vary the plant parameters for different crops (Root depths, crop coefficient, LAI, etc.)? Please name and quantify the parameters.

Chapter 4.2: You should not only the resulting streamflow pattern (Figure 6) but also the different ETA – for same HRUs (Hydrological response units) and for the entire catchment. How was the water balance changed?

Page 17, line 22: "In the Dinder River the effect of LULCC on streamflow is not big as in Rahad River." Please find reasons for this different behavior.
Conclusions: Please analyze also the effect of different precipitation patterns and magnitudes on streamflow in different years (2001 until 2012). When do you see a larger effect of LULCC on streamflow alteration? Find explanations for that.

---

## Author Comment (AC1) · 26 Jun 2017

hess-2017-128

**Author's response to reviewer's comments**

First, the authors would like to thank the editor Dr. Uwe Ehret for handling this manuscript as well as the two anonymous reviewers for their critical and constructive comments. The reviewer's comments and suggestions were highly insightful and enabled us to greatly improve the quality of the manuscript.

Here we present our response (in blue color) to all points raised during the review process.

**Anonymous Referee #1**

"The rainfall runoff process over the upper Dinder and Rahad basins (D&R) is complex, non-linear, and exhibits temporal and spatial variability". This needs a citation to support it.

The citation is made, and the sentence is rewritten as in the text below:

*"The rainfall runoff process over the upper Dinder and Rahad basins (D&R) is complex, non-linear, and exhibits temporal and spatial variability (Hassaballh et.al. 2016)".*

"However, the impact is often not well understood with locally obtained data such as observed flow". !The main issue here is the fact that data from local stations, is often not long enough or have periods of missing gaps. As it reads, it may look like the information from local stations is not understood.

This sentence is rewritten in the manuscript to make the connection between LULC changes and hydrological response downstream as monitored at the discharge stations.

*"However, the impact is often not well understood with only locally obtained data such as observed flow. This needs to be linked to the drivers which are likely affect the hydrological processes, such as the land use and land cover changes)."*

Citation in line should follow the author–date.

Agreed and corrected

**Reviewer's supplement to his comments:**

Page2 line 11: "has" corrected to "have".

Page2 line 14: (Hassaballah et al., 2016) corrected to "*Hassaballah et al., (2016)*".

Page3 line 12: here we referred to the Dinder and Rahad rivers. The sentence was corrected to "*Their catchments areas are about 34,964 and 42,540 km$^2$ for the Dinder and the Rahad, respectively, giving a total area of about 77,504 km$^2$.*"

In Figure 4.b, the reviewer is asking why the pattern of REF 2.0 during the first period of the validation period is different from the other two products. The response to this question is as follows:

The different pattern of RFE 2.0 in 2007 is likely due to an error in rainfall estimation by the RFE 2.0 product (please refer to Figure 4(new)).

**Anonymous Referee #2**
However, the paper needs improvement before getting published. I will not correct some English grammar / expressions.

A final proofreading from a native speaker is still recommended.
The English language of the manuscript has been improved

The comments are split into two parts:

1. Remote sensing:
Page 5 Line 9: The exact dates for these four years are needed. This is very important information for a land cover change detection analysis, also a brief information (e.g. coverage of cloud) about the quality of selected Landsat data is also necessary.

The Table below (Table 2new) is added to the manuscript in section 3.2 to give further information of the satellite data. Table 2 is anew Table, thus tables in the manuscript are updated accordingly.

Table 2: Description of used satellite images.

| Acquisition date | Satellite | Number of scenes | Spectral bands | Spatial resolution |
|---|---|---|---|---|
| 04 Nov. & 11 Dec. 1972 | Landsat MSS | 9 | 1 – 4 bands | 60 m |
| 12 Nov. & 26 Nov. 1986 | Landsat TM | 9 | 1 – 6 bands | 30 m |
| 27 Nov. & 13 Dec. 1998 | Landsat TM | 8 | 1 – 6 bands | 30 m |
| 07 Nov. & 10 Dec. 2011 | Landsat TM | 8 | 1 – 6 bands | 30 m |

MSS, multispectral scanner; TM, thematic mapper

In addition, the following explanation is added to the manuscript text (page5, section 3.2, line 9):

"*All acquired images had less than 10% cloud cover. However, in order to cover the entire study area, more than one scene of the satellite data was obtained (see Table 2new). Consequently, all images were mosaicked and resampled to a pixel size of 30m x 30m.*"

Page 5 Line 29. This description is not right. TMPA is just a product of TRMM. There are many TRMM products, here please specify which one you used. I think it should

be TRMM 3B42V7.

The reviewer is right; we have corrected the product to TRMM 3B42v7.

Page 5 Line 34: here it was described as CHIRPS available from 1981, but in the Table 1 it was mistakenly written from 1983. Additionally, CHRIPS provides daily data for the globe, please correct "Pentads" in Table 1 accordingly.

Table 1, CHIRPS availability corrected to 1981 instead of 1983, and "pentads" corrected to "daily".

Page 5 Line 36: The "TRMM" should be corrected to "CHIRPS".

"TRMM" corrected to "CHIRPS".

Table 1: These products have different spatial resolutions, the authors should explain how they processed such data (how to deal with the difference in spatial resolution) and used them as input to the model.

All RS data were projected into WGS-84-UTM-zone 36N (meters), clipped to the catchment extent, and resampled to a horizontal resolution of 500 m. The text below is inserted in section 3.1.2 of the manuscript to clarify spatial resolution issue further.

*"All maps were projected into WGS-84-UTM-zone 36N (meters), clipped to catchment extent, and then resampled to a resolution of 500 m".*

Figure 3: In the caption, "19986" should be "1986". The legend is quite abnormal, in the remote sensing analysis, crop is more commonly assigned to yellow color, while natural vegetation to green. I advise to change legend. The four land cover maps in Figure 3 shows quite remarked differences, and it seems no regular pattern, which needs more discussion and analysis about the quality of classified map. Normally there should be a pattern, because human activities follow rivers to convert natural vegetation to crop lands. I advise to use one or more matchup Google Earth High Resolution Images to further prove/evaluate the reliability of classified map.

The date "19986" corrected to 1986.

The legend is changed and the LULC color schematic is made clearer (see Figure 3 below).

It seems that the unclear patterns of LULC in the maps is due to the small-scale of the maps (i.e. 1:4,500,000), which may not allow distinction of different LULC change patterns by eye. Therefore, we have added Figure 3 modified, which zoom into smaller area as an example to show multi-temporal changes in the LULC patterns.

We have added the text below to the manuscript in section 4.1, to show a clear example of LULC multi-temporal change.

*"Due to the large extent of the catchment (77,504 km$^2$), the small-scale of the maps (i.e. 1:4,500,000) that covered the whole catchment, may not allow distinction of different LULC change patterns by eye. Therefore, we have added Figure 3 modified, which zoom into smaller area as an example to show multi-temporal changes in the LULC patterns. The zoomed areas in the red boxes showed in large-scale to provide more details of LULC patterns. This area is located down to the Rahad Irrigation Scheme of Sudan conducted in 1981. Therefore, the waterlogging and woodland areas occurred in 1998 and 2011 resulted from the drainage water of the project accumulated over the years (i.e. clear example of LULC multi-temporal change over the Rahad basin). The lowest maps indicate the Google Earth layers of the large-scale area. Although the Google Earth layers dates were not exactly match the ones of the satellite images, but it shows part of the dried period in the study area and hence the complexity of the LULC patterns."*

[Figure]

Figure 3: Classified LULC maps of the years 1972, 1986, 1998 and 2011.

[Figure]

**Figure 3_modiffed**: Classified LULC maps of the years 1972, 1986, 1998 and 2011. The areas in the red boxes showed in large-scale to provide more details of LULC patterns.

2. Hydrological Modelling:
Figure 1. I recommend to insert the Blue Nile River in the upper right map.

Agreed, Blue Nile River is inserted in the upper right map. Also, the rain gauges are added to this Figure (see Figure1_modified).

[Figure]

Figure 1_modified: Location map of the Dinder and Rahad basins and the DNP. The two black stars are the hydrological stations (Al-Gewisi and Al-Hawata). The red triangles are the rain gauges stations.

Page 5, line 15: Why did you reclassify the 44 soil mapping units into 8 dominant soil groups? Was this necessary for the hydrological modelling?

Yes, this was necessary to comply with parametrization of hydrological model Wflow. The reclassification of the 44 soil units into 8 dominant soil groups reduces the number of estimated parameters to 64 instead of 352. We have added this explanation to text on page 5 (section 3.1.1

*"This was done to reduce the model complexity. The WFlow soil model requires estimate of 8 parameters per soil type, which means 352 parameters for 44 soil types. Therefore, reclassification of soil map into 8 dominant soil groups reduces the number of estimated parameters to 64"*.

Figure 2: The WFlow_sbm model needs more explanation. How is runoff generation modelled? How is ETA calculated? Is there no interflow component?

We have added below text to further describe the Wflow model.

*"The model uses a series of expressions to calculate the interception loss. An analytical combination of the total rainfall and evaporation under condition of saturated canopy is done for each rainfall storm to determine average values of precipitation and evaporation from the wet canopy. In case the soil surface is partially saturated, the rainfall that falls on the saturated area is directly added to the surface runoff component. The soil is represented by a simple bucket model that assumes an exponential decay of the saturated conductivity with depth. Lateral subsurface flow is simulated using the Darcy equation. Soil depth is identified for different land use types and consequently scaled using the Topographic Wetness Index. The WFlow_sbm interception and soil model's equations are presented in Appendix A." Further details of Wflow model is also given at* [https://media.readthedocs.org/pdf/wflow/latest/wflow.pdf](https://media.readthedocs.org/pdf/wflow/latest/wflow.pdf).

Chapter 3.4: The IHA approach should be explained in more detail (add app. half page)

We have added below text to further describe the IHA approach.

*"The IHA technique computes 33 hydrologic parameters for each year. For analyzing the alteration between two periods, the IHA described in Richter et al. (1996) was applied using the IHA software developed by The Nature of Conservancy (2009).*

*The general approach is to define hydrologic parameters that characterized the intra-annual variation in water system condition and then to use the analysis of variations in these parameters as a base for comparing hydrologic alterations of the system before and after the system has been altered by various human activities.*

*The IHA method has four steps: a) define the time series of the hydrologic variable (e.g. streamflow) for the two periods to be compared; b) calculate values for hydrologic parameters; c) compute intra-annual statistics; and d) calculate values of the IHA by comparing the intra-annual variation before and after the system has been altered and present the results as a percentage of deviation. For assessing hydrologic alteration in the Dinder and Rahad Rivers, the flows variations for both rivers have been characterized based on the variations in the streamflow characteristics between three periods (1972-1986), (1986-1998) and (1998-2011). Temporal variability of streamflow series was analyzed at Al-Gewisi station on the Dinder River and at Al-Hawata station on the Rahad River."*

Table 2: Please explain the accuracy assessment. What means "producer" and "user"? Did you perform a cross validation analysis? The accuracy seems very high with little uncertainty for all classes. Can you prove this?

The accuracy assessment is based on comparing reference data (class types at specific locations from ground information) to image classification results at the same locations. The overall accuracy of classification is the average value from all classes. The User's accuracy corresponds to error of commission (inclusion). It represents the probability of a pixel classified into a given category actually represents that category on the ground (i.e. from the perspective of the user of the classified map, how accurate is the map?). The Producer's accuracy corresponds to error of omission (exclusion). It represents how well reference pixels of the ground cover type are classified (i.e. from the perspective of the maker of the classified map, how accurate is the map?).

The cross validation for the 2011 land use was made using the reference data (120 points) collected using the GPS instrument during the field survey (2011–2013) assuming no significant change during this period. In addition, visual interpretation and historical information obtained from the local indigenous about the land use types in the study area were used also as cross-check validations for old maps (please refer to the text, page 6: lines 10-27).

Below is an example of the classification error matrix for the image of the year 2011. The error of commission and omission varies from 75 to 94% for the different LULC classes

| Image Date: 2011 | | Class types determined from reference data | | | | | | |
|---|---|---|---|---|---|---|---|---|
| | LULC | Woodland | Shrub Land | Grass Land | Crop Land | Bare Land | Water | Row Total |
| Class types determined from classified map | Woodland | 130 | 3 | 2 | 2 | 1 | 2 | 140 |
| | Shrub Land | 4 | 36 | 4 | 4 | 0 | 0 | 48 |
| | Grass Land | 1 | 2 | 68 | 2 | 3 | 0 | 76 |
| | Crop Land | 2 | 3 | 3 | 45 | 2 | 0 | 55 |
| | Bare Land | 1 | 1 | 2 | 1 | 28 | 0 | 33 |
| | Water | 5 | 0 | 0 | 0 | 0 | 30 | 35 |
| | Column Total | 143 | 45 | 79 | 54 | 34 | 32 | 387 |
| Accuracies (%) | | | | | | | | |
| Producer's | | 91* | 80 | 86 | 83 | 82 | 94 | |
| User's | | 93** | 75 | 89 | 82 | 85 | 86 | |
| Overall | | 87 | | | | | | |

We have added the following text to the manuscript in section 4.1 to give further clarification on accuracy

*"The User's accuracy indicates the accuracy of the classification from the user perspective, while the producer's accuracy shows the map maker perspective of the classified map. Shrub lands show lower user's and producer's accuracies compared to the other LULC classes. This is mainly due to the miss-classification of some shrub land pixels, which been classified as woodland, grassland and cropland."*

Chapter 4.1.1 / Figure 4 and Figure 5: A critical discussion about the calibration and its uncertainties is totally missing. Couldn't you assess the reliability of the RS data by ground truth measurements (rain gauges)? Please comment on that. Concerning the figures, there are great differences in the peak flows with reverse biases. For instance, at Al-Gewisi station, you get a large underestimation in the first validation period for CHIRPS, whereas you get a large underestimation for the same time period and RS method for the Al-Hawata station. There are many contrary results comparing the two figures. Please discuss this issue. Moreover, did you vary the plant parameters for different crops (Root depths, crop coefficient, LAI, etc.)? Please name and quantify the parameters.

Our response to this comment is divided into three parts as follows:

Chapter 4.1.1 / Figure 4 and Figure 5: A critical discussion about the calibration and its uncertainties is totally missing. Couldn't you assess the reliability of the RS data by ground truth measurements (rain gauges)? Please comment on that.

We must admit that we haven't conducted an in-depth validation of the Satellite based rainfall Estimate (SBRE) products. It is beyond the scope of this manuscript. However, we assessed the reliability of SBRE products by direct comparison against ground measurements at four locations. The results are shown in Figure 4new.

We have inserted below text including Figure 4 new in section 4.1.1 of the manuscript

*"To assess the reliability of the SBRE products, validation is done with the use of ground measurements at four gauges in which observed data are available. Two gauges (Gonder and Bahir Dar) are located nearby the upstream part of the catchments in the Ethiopian plateau, while the others two (Gedarif and Al-Hawata) are located at the most downstream part of the catchment in the Sudan low land. The validation is performed at annual time step. The results show that the difference of RFE against ground measurements has no consistent patterns. TRMM and CHIRPS have shown no consistent patterns at the lowland (Gedarif and Al-Hawata), but both products are consistent and overestimate rainfall at the Ethiopian highland (Gonder and Bahir Dar) in all years except 2007 (Figure 4). Since both the Dinder and Rahad derive their main flow from the Ethiopian highlands, products with consistent patterns at highlands will be more suitable for running hydrologic models in this catchment. From these findings, one can conclude that the CHIRPS v2.0 and TRMM 3B42 v7 are more suitable than RFE 2.0 for running hydrologic model. Comparing CHIRPS v2.0 and TRMM 3B42 v7, it is clear that CHIRPS v2.0 has less overestimation of rainfall. Thus, CHIRPS v2.0 is the best product to be used as a forcing data for hydrologic model in the Dinder and Rahad Basins".*

[Figure]

Figure 4new: Comparison of SBRE products with ground measurements at four locations

Concerning the figures, there are great differences in the peak flows with reverse biases. For instance, at Al-Gewisi station, you get a large underestimation in the first validation period for CHIRPS, whereas you get a large underestimation for the same time period and RS method for the Al-Hawata station. There are many contrary results comparing the two figures. Please discuss this issue.

The differences in the peak flow with reverse bias can be attributed to the differences in rainfall estimation with different products (see Figure 4new). At Al-Gewisi station, the large underestimation in the first validation period for CHIRPS can be attributed to the underestimation of rainfall by CHIRPS in 2007 at both Gonder and Bahir Dar (Figure 4 new). At the same time CHIRPS overestimates rainfall in all years from 2001 to 2006. Therefore, calibration of the hydrologic model based on (2002-2005) will result in a more underestimation of river flow for the year 2007. This is confirmed by Figure 4 in the original manuscript and supported by the validation result in Figure 4new presented in this document. On the other hand, at Al-Hawata station the difference between observed and model flow in the first period of validation (2008) is difficult to explain in terms of problem in model structure as it is a single event out of the 4 years validation period. It can likely be due to error either in the input data or the observed flow or a combination of

both. Results of the three SBRE products at Al-Gewisi and Al-Hawata (Figure 4 and 5) in the original manuscript are consistent and supported by the validation results in Figure 4new. For instance, in 2004 the model results for RFE 2.0 underestimate the flow at both Al-Gewisi and Al-Hawata stations. This is supported by the rainfall validation results in Figure 4new which shows that RFE 2.0 underestimates rainfall in all stations. Also in 2010, the model results for TRMM v7 underestimate the flow at both Al-Gewisi and Al-Hawata stations. This is also supported by the validation results in Figure 4new which shows that TRMM v7 underestimates rainfall in Gedarif station. The text hereafter is added to section 4.1.1 line 12.

*"At Al-Gewisi station, the large underestimation in the first validation period for CHIRPS can be attributed to the underestimation of rainfall by CHIRPS in 2007 at both Gonder and Bahir Dar (see Figure 4). At the same time CHIRPS overestimates rainfall in all years from 2001 to 2006. Therefore, calibration of the hydrologic model based on (2002-2005) resulted in a more underestimation of river flow in 2007. On the other hand, at Al-Hawata station, the difference between observed and model flow in the first period of validation (i.e.2008) is difficult to explain in terms of problem in model structure as it is a single event out of the 4 years validation period. It can likely be due to error either in the input data or the observed flow or a combination of both."*

Moreover, did you vary the plant parameters for different crops (Root depths, crop coefficient, LAI, etc.)? Please name and quantify the parameters.

Yes, indeed each vegetation cover is assigned different parameters when modeling the flow response to different land cover. These parameters include; root depth, leaf area index (LAI), evaporation from wet canopy/average rainfall (E/R) ratio, Albedo, Canopy Gap Fraction and Maximum Canopy Storage. All model parameters are linked to the Wflow model through lookup tables. The text below is added to the manuscript.

*"Different parameters are assigned to each land cover type. These parameters include; rooting depth, leaf area index (LAI), ratio of evaporation from wet canopy to average rainfall ($E_w/R$), Albedo, Canopy Gap Fraction and Maximum Canopy Storage. All model parameters are linked to the Wflow model through lookup tables. The lookup tables are used by the model to create input parameter maps. Each table consists of four columns. The first column is used to identify the land-use class, the second column indicates the sub catchment, the third column represents the soil type and the last column lists the assigned values based on the first three columns. The parameters are linked to land use, soil type or sub-catchment through lookup tables. Some parameters have a single value and not linked to any of the maps. Description of the Wflow model parameters is presented in Appendix B and the calibrated values for each parameter are presented in Appendix C."*

Chapter 4.2: You should not only the resulting streamflow pattern (Figure 6) but also the different ETA – for same HRUs (Hydrological response units) and for the entire

catchment. How was the water balance changed?

In Figure 6, we have shown the effect of LULCC on the daily streamflow (peak flows). However, to show the changes in AET at HRUs (sub-catchments) and the water balance for the entire catchment, the discussion in the text and the new tables (Table 4 and 5) below are added to section 4.2.

*"In addition to the streamflow response to LULCC, Evapotranspiration (ET) is another important component of the water balance that constitutes a major determinant of the amounts of water draining from different land cover type within the catchment. The ET result shows high rates of actual evapotranspiration (AET) when running the model with land cover of 1972 and 1998 at both the sub-catchments and the entire catchment (Table 4 and 5) . This can be attributed to the large percentage coverage of woodland in 1972 and 1998 compared to land cover of 1986 and 2011 (please refer to table 3 in the main manuscript). The lowest AET is observed when running the model with land cover of 1986. This is likely due to the severe drought during the mid-1980s that limits the water availability and decreases the green coverage. Table 4 presents the changing in the annual average AET at sub-catchment level as a response to LULCC for the Dinder catchment. Table 5 shows the changes in water balance for the entire Dinder and Rahad catchments when running the hydrologic model with different LULC and fixed rainfall data for the periods (2001-2012)."*

Table 4: Annual average AET as a response to LULCC at the sub-catchments level for the Dinder catchment (1972-1986).

| | AET from land cover of 1972 | | | | AET from land cover of 1986 | | | |
|---|---|---|---|---|---|---|---|---|
| Year | Al-Gewisi | Musa | Gelagu | Upper Dinder | Al-Gewisi | Musa | Gelagu | Upper Dinder |
| 2001 | 558 | 583 | 626 | 464 | 426 | 424 | 396 | 288 |
| 2002 | 443 | 456 | 535 | 510 | 322 | 317 | 306 | 312 |
| 2003 | 564 | 639 | 642 | 486 | 425 | 469 | 405 | 312 |
| 2004 | 455 | 502 | 573 | 500 | 326 | 354 | 340 | 311 |
| 2005 | 504 | 547 | 575 | 505 | 376 | 396 | 358 | 323 |
| 2006 | 527 | 576 | 632 | 545 | 396 | 414 | 406 | 359 |
| 2007 | 598 | 602 | 618 | 564 | 468 | 444 | 400 | 382 |
| 2008 | 593 | 689 | 703 | 576 | 459 | 513 | 471 | 392 |
| 2009 | 421 | 482 | 519 | 516 | 310 | 343 | 302 | 323 |
| 2010 | 536 | 566 | 606 | 520 | 412 | 415 | 383 | 331 |
| 2011 | 470 | 467 | 554 | 530 | 350 | 327 | 329 | 332 |
| 2012 | 636 | 679 | 684 | 542 | 500 | 504 | 450 | 353 |

Table 5: Water balance of the Dinder and Rahad catchments as a response to LULCC

| Dinder catchment | | Land cover of 1972 | | Land cover of 1986 | | Land cover of 1998 | | Land cover of 2011 | |
|---|---|---|---|---|---|---|---|---|---|
| Year | Rainfall | AET | Streamflow | AET | Streamflow | AET | Streamflow | AET | Streamflow |
| 2001 | 816 | 558 | 258 | 383 | 433 | 432 | 384 | 496 | 320 |
| 2002 | 663 | 486 | 177 | 314 | 349 | 364 | 299 | 430 | 233 |
| 2003 | 847 | 583 | 264 | 403 | 444 | 449 | 397 | 519 | 327 |
| 2004 | 703 | 507 | 195 | 333 | 370 | 374 | 329 | 451 | 252 |
| 2005 | 768 | 532 | 236 | 363 | 405 | 414 | 354 | 479 | 289 |
| 2006 | 835 | 570 | 265 | 394 | 441 | 441 | 395 | 513 | 322 |
| 2007 | 876 | 595 | 280 | 424 | 452 | 476 | 400 | 540 | 336 |
| 2008 | 929 | 640 | 289 | 459 | 470 | 509 | 420 | 582 | 347 |
| 2009 | 659 | 484 | 175 | 319 | 340 | 363 | 297 | 435 | 225 |
| 2010 | 817 | 557 | 260 | 385 | 432 | 432 | 386 | 505 | 312 |
| 2011 | 710 | 505 | 205 | 334 | 376 | 377 | 333 | 454 | 256 |
| 2012 | 972 | 635 | 337 | 452 | 520 | 498 | 474 | 579 | 393 |
| Rahad catchment | | Land cover of 1972 | | Land cover of 1986 | | Land cover of 1998 | | Land cover of 2011 | |
| Year | Rainfall | AET | Streamflow | AET | Streamflow | AET | Streamflow | AET | Streamflow |
| 2001 | 724 | 409 | 315 | 290 | 434 | 398 | 326 | 309 | 416 |
| 2002 | 641 | 398 | 243 | 271 | 370 | 383 | 258 | 291 | 350 |
| 2003 | 755 | 450 | 305 | 323 | 432 | 434 | 322 | 342 | 413 |
| 2004 | 609 | 360 | 249 | 231 | 378 | 338 | 270 | 244 | 364 |
| 2005 | 656 | 399 | 258 | 267 | 389 | 378 | 278 | 285 | 372 |
| 2006 | 782 | 450 | 332 | 324 | 457 | 431 | 351 | 336 | 446 |
| 2007 | 774 | 473 | 301 | 344 | 430 | 456 | 319 | 363 | 411 |
| 2008 | 754 | 438 | 315 | 313 | 441 | 415 | 338 | 322 | 431 |
| 2009 | 581 | 352 | 229 | 220 | 361 | 333 | 248 | 238 | 343 |
| 2010 | 744 | 449 | 295 | 319 | 425 | 431 | 313 | 335 | 409 |
| 2011 | 610 | 369 | 241 | 235 | 375 | 348 | 262 | 252 | 358 |
| 2012 | 873 | 507 | 366 | 381 | 492 | 485 | 388 | 390 | 483 |

Page 17, line 22: "In the Dinder River the effect of LULCC on streamflow is not big as in Rahad River." Please find reasons for this different behavior.

The reason for this was mentioned in the conclusion, line 20. However, for more clarity the explanation in the text below is added to section 4.3.2, Page 17, lines 22-23.

*"This is likely due to the large expansion in cropland in the Rahad catchment to 68% of the total area compared to 47% in the Dinder catchment".*

Conclusions: Please analyze also the effect of different precipitation patterns and magnitudes on streamflow in different years (2001 until 2012). When do you see a larger effect of LULCC on streamflow alteration? Find explanations for that.

Replying to the first part of this comment "Please analyze also the effect of different precipitation patterns and magnitudes on streamflow in different years (2001 until 2012)", the below coated text in addition to the new figure will be added to section 4.2.

*"Since both Dinder and Rahad rivers are seasonal, their flows are mainly depending on rainfall patterns and magnitudes. In addition to the effect of LULCC on the streamflow, Figure 8(new) shows that the annual variability of rainfall is another factor affecting the annual patterns of the streamflow." Since Figure 8new is a new Figure, figures in the manuscript are updated accordingly."*

[Figure]

Figure 8: Annual rainfall and streamflow patterns and magnitudes for the years (2001-2012).

Our reply to the second part of the above comment "When do you see a larger effect of LULCC on streamflow alteration? Find explanations for that.", is as follows:

The effect of LULCC on streamflow is found to be large when running the model with land use of 1986 
[revised manuscript text omitted]

|---|---|---|---|
| 1 | [0,> | [0,> | 0.40 |
| 2 | [0,> | [0,> | 0.20 |
| 3 | [0,> | [0,> | 0.16 |
| 4 | [0,> | [0,> | 0.26 |
| 5 | [0,> | [0,> | 0.25 |
| 6 | [0,> | [0,> | 0.10 |

CanopyGapFraction

| Land cover | Sub-catchment | Soil class | Value |
|---|---|---|---|
| 1 | [0,> | [0,> | 1.0 |
| 2 | [0,> | [0,> | 0.2 |
| 3 | [0,> | [0,> | 0.6 |
| 4 | [0,> | [0,> | 0.5 |
| 5 | [0,> | [0,> | 0.4 |
| 6 | [0,> | [0,> | 0.5 |

EoverR

| Land cover | Sub-catchment | Soil class | Value |
|---|---|---|---|
| 1 | [0,> | [0,> | 0.0 |
| 2 | [0,> | [0,> | 0.3 |
| 3 | [0,> | [0,> | 0.2 |
| 4 | [0,> | [0,> | 0.2 |
| 5 | [0,> | [0,> | 0.1 |
| 6 | [0,> | [0,> | 0.0 |

FirstZoneCapacity

| Land cover | Sub-catchment | Soil class | Value |
|---|---|---|---|
| [0,> | [0,> | 1 | 44500 |
| [0,> | [0,> | 2 | 42000 |
| [0,> | [0,> | 3 | 44500 |
| [0,> | [0,> | 4 | 39000 |
| [0,> | [0,> | 5 | 44000 |
| [0,> | [0,> | 6 | 42000 |
| [0,> | [0,> | 7 | 44500 |

FirstZoneKsatVer

| Land cover | Sub-catchment | Soil class | Value |
|---|---|---|---|
| [0,> | [0,> | 1 | 511 |
| [0,> | [0,> | 2 | 600 |
| [0,> | [0,> | 3 | 543 |
| [0,> | [0,> | 4 | 525 |
| [0,> | [0,> | 5 | 586 |
| [0,> | [0,> | 6 | 576 |
| [0,> | [0,> | 7 | 540 |

FirstZoneMinCapacity

| Land cover | Sub-catchment | Soil class | Value |
|---|---|---|---|
| [0,> | [0,> | 1 | 125 |
| [0,> | [0,> | 2 | 50 |
| [0,> | [0,> | 3 | 137.5 |
| [0,> | [0,> | 4 | 33 |
| [0,> | [0,> | 5 | 87.5 |
| [0,> | [0,> | 6 | 60 |
| [0,> | [0,> | 7 | 70 |

InfiltCapPath

| Land cover | Sub-catchment | Soil class | Value |
|---|---|---|---|
| [0,> | [0,> | 1 | 5 |
| [0,> | [0,> | 2 | 21 |
| [0,> | [0,> | 3 | 5 |
| [0,> | [0,> | 4 | 32 |
| [0,> | [0,> | 5 | 34 |
| [0,> | [0,> | 6 | 5 |
| [0,> | [0,> | 7 | 21 |

InfiltCapSoil

| Land cover | Sub-catchment | Soil class | Value |
|---|---|---|---|
| [0,> | [0,> | 1 | 24 |
| [0,> | [0,> | 2 | 103 |
| [0,> | [0,> | 3 | 24 |
| [0,> | [0,> | 4 | 158 |
| [0,> | [0,> | 5 | 170 |
| [0,> | [0,> | 6 | 100 |
| [0,> | [0,> | 7 | 103 |

LeafAreaIndex

| Land cover | Sub-catchment | Soil class | Value |
|---|---|---|---|
| 1 | [0,> | [0,> | 0.0 |
| 2 | [0,> | [0,> | 8.8 |
| 3 | [0,> | [0,> | 7.0 |
| 4 | [0,> | [0,> | 0.6 |
| 5 | [0,> | [0,> | 0.7 |
| 6 | [0,> | [0,> | 0.0 |

M

| Land cover | Sub-catchment | Soil class | Value |
|---|---|---|---|
| [0,> | [0,> | 1 | 100 |
| [0,> | [0,> | 2 | 87 |
| [0,> | [0,> | 3 | 100 |
| [0,> | [0,> | 4 | 77 |
| [0,> | [0,> | 5 | 100 |
| [0,> | [0,> | 6 | 100 |
| [0,> | [0,> | 7 | 100 |

MaxCanopyStorage

| Land cover | Sub-catchment | Soil class | Value |
|---|---|---|---|
| 1 | [0,> | [0,> | 0.00 |
| 2 | [0,> | [0,> | 0.336 |
| 3 | [0,> | [0,> | 0.21 |
| 4 | [0,> | [0,> | 0.25 |
| 5 | [0,> | [0,> | 0.34 |

N

| Land cover | Sub-catchment | Soil class | Value |
|---|---|---|---|
| 1 | [0,> | [0,> | 0.42 |
| 2 | [0,> | [0,> | 0.80 |
| 3 | [0,> | [0,> | 0.70 |
| 4 | [0,> | [0,> | 0.65 |
| 5 | [0,> | [0,> | 0.80 |

| 6 | [0,> | [0,> | 0.00 |
|---|------|------|------|

**PathFrac**

| Land cover | Sub-catchment | Soil class | Value |
|------------|---------------|------------|-------|
| [0,> | [0,> | 1 | 0.06 |
| [0,> | [0,> | 2 | 0.09 |
| [0,> | [0,> | 3 | 0.05 |
| [0,> | [0,> | 4 | 0.06 |
| [0,> | [0,> | 5 | 0.06 |
| [0,> | [0,> | 6 | 0.07 |
| [0,> | [0,> | 7 | 0.08 |

**thetaR**

| Land cover | Sub-catchment | Soil class | Value |
|------------|---------------|------------|-------|
| [0,> | [0,> | 1 | 0.15 |
| [0,> | [0,> | 2 | 0.09 |
| [0,> | [0,> | 3 | 0.19 |
| [0,> | [0,> | 4 | 0.09 |
| [0,> | [0,> | 5 | 0.11 |
| [0,> | [0,> | 6 | 0.09 |
| [0,> | [0,> | 7 | 0.08 |

**RunoffGeneratingGWPerc**

| Land cover | Sub-catchment | Soil class | Value |
|------------|---------------|------------|-------|
| [0,> | [0,> | [0,> | 0.1 |

**N_River**

| Land cover | Sub-catchment | Soil class | Value |
|------------|---------------|------------|-------|
| [0,> | [0,> | [0,> | 0.035 |

**CapScale**

| Land cover | Sub-catchment | Soil class | Value |
|------------|---------------|------------|-------|
| [0,> | [0,> | [0,> | 100 |

| 6 | [0,> | [0,> | 0.12 |
|---|------|------|------|

**RootingDepth**

| Land cover | Sub-catchment | Soil class | Value |
|------------|---------------|------------|-------|
| 1 | [0,> | [0,> | 1000 |
| 2 | [0,> | [0,> | 1800 |
| 3 | [0,> | [0,> | 1400 |
| 4 | [0,> | [0,> | 1600 |
| 5 | [0,> | [0,> | 200 |
| 6 | [0,> | [0,> | 0 |

**thetaS**

| Land cover | Sub-catchment | Soil class | Value |
|------------|---------------|------------|-------|
| [0,> | [0,> | 1 | 0.5 |
| [0,> | [0,> | 2 | 0.2 |
| [0,> | [0,> | 3 | 0.5 |
| [0,> | [0,> | 4 | 0.3 |
| [0,> | [0,> | 5 | 0.4 |
| [0,> | [0,> | 6 | 0.2 |
| [0,> | [0,> | 7 | 0.2 |

**rootdistpar**

| Land cover | Sub-catchment | Soil class | Value |
|------------|---------------|------------|-------|
| [0,> | [0,> | [0,> | -80000 |

**Beta**

| Land cover | Sub-catchment | Soil class | Value |
|------------|---------------|------------|-------|
| [0,> | [0,> | [0,> | 0.6 |

Land cover: 1= Bare land, 2= woodland, 3= shrubland, 4= grassland, 5= cropland, 6= water bodies.
Soil type: 1= Vertisols, 2= Luvisols, 3= Nitisols, 4= Leptosols, 5= cambisols, 6= Alisols, 7= Fluvisols.

---

## Author Response (AR1)

**Author's response to editor and reviewer's comments**

First, the authors would like to thank the editor Dr. Uwe Ehret for handling this manuscript as well as the two anonymous reviewers for their critical and constructive comments. The editor's and reviewer's comments and suggestions were highly insightful and enabled us to greatly improve the quality of the manuscript.

Here we present our response (in blue color) to all points raised during the review process and added the revised marked-up version of the manuscript after the replies. The revised manuscript is also uploaded.

**Editor's comments:**

**Comment:** The points raised by the referees have been suitably addressed in your replies. Please change your manuscript accordingly.

**Response:**
The manuscript has been revised according to the comments of the anonymous referees and the editor (please see both the revised manuscript and the marked-up manuscript version).

**Comment:** In addition, please:
- change the manuscript according to the comment by referee #2 (page 5 line 29): 'This description is not right. TMPA is just a product of TRMM'. This has not been addressed in your reply.

**Response:**
The referee #2 is right, TMPA is a product of TRMM. The sentence "TRMM product uses a multi-satellite precipitation analysis (TMPA), which includes also ground measurements provided by the Global Precipitation Climatology Center (GPCC)" is replaced by: "*The TRMM satellite rainfall measuring instruments include the Precipitation Radar (PR), TRMM Microwave Image (TMI), a nine-channel passive microwave radiometer, a Visible and Infrared Scanner (VIRS), and five-channel visible/infrared radiometer (Huffman and Bolvin 2013)*". We have also modified our response to the referee #2 regarding this comment.

**Comment:** comment of referee #2 about table 2: This point is indeed hard to understand and I suggest you add to the manuscript an in-depth explanation (comparable to your reply to the referee), instead of the relatively short text you proposed to add.

**Response:**
Agreed. We have added the following text to the manuscript in section 4.1 to give further clarification on accuracy assessment.

*"The accuracy assessment is based on comparing reference data (class types at specific locations from ground information) to image classification at the same locations. The overall accuracy of classification is the average value from all classes. The user's accuracy corresponds to errors of inclusion (commission errors), which represents the probability of a pixel classified into a given class actually represents that class on the ground (i.e. from the perspective of the user of the classified map). The producer's accuracy corresponds to errors of exclusion (omission errors), which represents how well reference pixels of the land cover type are classified (i.e. from the perspective of the maker of the classified map). The commission errors occur when an area is included in an incorrect category, while the omission errors occur when an area is excluded from the category to which it belongs. Every error on the map is an omission from the correct class and a commission to an incorrect class (Congalton and Green 2008). The cross validation for the year 2011 land use was made using the reference data (120 points) collected with GPS instrument during the field survey (2011–2013). In addition, visual interpretation and historical information obtained from the local people about the land use types in the study area were used also as cross-check validations for old maps. Shrublands show lower user's and producer's accuracies compared to the other LULC classes. This is mainly due to the miss-classification of some shrub land into woodland, grassland and cropland."*

Notes:
- Page and line numbers in the reviewer's comments refer to page and line numbers in the manuscript-version1.
- Page and line numbers in the author's replies refer to page and line numbers in the revised marked-up manuscript.

**Anonymous Referee #1**

**Comment:** "The rainfall runoff process over the upper Dinder and Rahad basins (D&R) is complex, non-linear, and exhibits temporal and spatial variability". This needs a citation to support it.

**Response:**
The citation is made, and text modified as:

*"The rainfall runoff process over the upper Dinder and Rahad basins (D&R) is complex, non-linear, and exhibits temporal and spatial variability (Hassaballah et.al. 2016)".*

**Comment:** "However, the impact is often not well understood with locally obtained data such as observed flow". !The main issue here is the fact that data from local stations, is often not long enough or have periods of missing gaps. As it reads, it may look like the information from local stations is not understood.

**Response:**
This sentence is rewritten in the manuscript to make the connection between LULC changes and hydrological response downstream as monitored at the discharge stations.

*"Therefore, it is necessary to understand the hydrological processes in the run-off generated catchments, and the possible interlinkages of land use and land cover changes with catchment runoff."*

**Comment:** Citation in line should follow the author–date.

**Response:**
Agreed and corrected

**Reviewer's supplement to his comments:**

Page2 line 11: "has" corrected to "have".

Page2 line 14: (Hassaballah et al., 2016) corrected to "*Hassaballah et al., (2016)*".

Page3 line 12: here we referred to the Dinder and Rahad rivers. The sentence was corrected to "*Their catchments areas are about 34,964 and 42,540 $km^2$ for the Dinder and the Rahad, respectively, giving a total area of about 77,504 $km^2$.*"

**Comment:** In Figure 4.b, the reviewer is asking why the pattern of REF 2.0 during the first period of the validation period is different from the other two products. The response to this question is as follows:

**Response:**
The different pattern of RFE 2.0 in 2007 is likely due to an error in rainfall estimation by the RFE 2.0 product (please refer to Figure 4(new)).

**Anonymous Referee #2**
**Comment:** However, the paper needs improvement before getting published. I will not correct some English grammar / expressions.  A final proofreading from a native speaker is still recommended.

**Response:**
The English language of the manuscript has been improved

**Comment:** The comments are split into two parts:

1. Remote sensing:

**Comment:** Page 5 Line 9: The exact dates for these four years are needed. This is very important information for a land cover change detection analysis, also a brief information (e.g. coverage of cloud) about the quality of selected Landsat data is also necessary.

**Response:**
The Table below (Table 2new) is added to the manuscript in section 3.2 to give further information of the satellite data. Table 2 is a new Table, thus tables in the manuscript are updated accordingly.

Table 2new: Description of used satellite images.

| Acquisition date | Satellite | Number of scenes | Spectral bands | Spatial resolution |
|---|---|---|---|---|
| 04 Nov. & 11 Dec. 1972 | Landsat MSS | 9 | 1 – 4 bands | 60 m |
| 12 Nov. & 26 Nov. 1986 | Landsat TM | 9 | 1 – 6 bands | 30 m |
| 27 Nov. & 13 Dec. 1998 | Landsat TM | 8 | 1 – 6 bands | 30 m |
| 07 Nov. & 10 Dec. 2011 | Landsat TM | 8 | 1 – 6 bands | 30 m |

MSS, multispectral scanner; TM, thematic mapper

In addition, the following explanation is added to the manuscript text (page 6, section 3.2, lines 15-16):

"*All acquired images had less than 10% cloud cover. However, in order to cover the entire study area, more than 8 scenes of the satellite data were processed (Table 2). Subsequently, all images were mosaicked and resampled to a pixel size of 30m x 30m*"

**Comment:** Page 5 Line 29. This description is not right. TMPA is just a product of TRMM. There are many TRMM products, here please specify which one you used. I think it should be TRMM 3B42V7.

**Response:**

The reviewer is right, TMPA is a product of TRMM. The sentence "TRMM product uses a multi-satellite precipitation analysis (TMPA), which includes also ground measurements provided by the Global Precipitation Climatology Center (GPCC)" is replaced by: "*The TRMM satellite rainfall measuring instruments include the Precipitation Radar (PR), TRMM Microwave Image (TMI), a nine-channel passive microwave radiometer, a Visible and Infrared Scanner (VIRS), and five-channel visible/infrared radiometer (Huffman and Bolvin 2013)*". The used product was corrected to TRMM 3B42v7.

**Comment:** Page 5 Line 34: here it was described as CHIRPS available from 1981, but in the Table 1 it was mistakenly written from 1983. Additionally, CHRIPS provides daily data for the globe, please correct "Pentads" in Table 1 accordingly.

**Response:**

Table 1, CHIRPS availability corrected to 1981 instead of 1983, and "pentads" corrected to "daily".

**Comment:** Page 5 Line 36: The "TRMM" should be corrected to "CHIRPS".

**Response:**

"TRMM" corrected to "CHIRPS".

**Comment:** Table 1: These products have different spatial resolutions, the authors should explain how they processed such data (how to deal with the difference in spatial resolution) and used them as input to the model.

**Response:**

All RS data were projected into WGS-84-UTM-zone 36N (meters), clipped to the catchment extent, and resampled to a horizontal resolution of 500 m. The text below is inserted in section 3.1.2 of the manuscript to clarify spatial resolution issue further.

*"All maps were projected into WGS-84-UTM-zone 36N (meters), clipped to catchment extent, and then resampled to a resolution of 500 m".*

**Comment:** Figure 3: In the caption, "19986" should be "1986". The legend is quite abnormal, in the remote sensing analysis, crop is more commonly assigned to yellow color, while natural vegetation to green. I advise to change legend. The four land cover maps in Figure 3 shows quite remarked differences, and it seems no regular pattern, which needs more discussion and analysis about the quality of classified map. Normally there should be a pattern, because human activities follow rivers to convert natural vegetation to crop lands. I advise to use one or more matchup Google Earth High Resolution Images to further prove/evaluate the reliability of classified map.

**Response:**

The date "19986" corrected to 1986.

**Response:**

The legend is changed and the LULC color schematic is made clearer (see Figure 3_modified).

**Response:**

It seems that the unclear patterns of LULC in the maps is due to the small-scale of the maps (i.e. 1:4,500,000), which may not allow distinction of different LULC change patterns by eye. Therefore, we have added a new figure (Figure 4), which zoom into smaller area as an example to show multi-temporal changes in the LULC patterns.

**Response:**

We have added the text below to the manuscript in section 4.1, to show a clear example of LULC multi-temporal change.
*"The large extent of the catchment (77,504 km2), and the small-scale of the maps (i.e. 1: 4,500,000), may not allow distinction of different LULC change patterns by eye. Figure 4 which zoom into smaller area is an example to show multi-temporal changes in the LULC patterns. The zoomed areas in the red boxes showed in large-scale, provides more details of LULC patterns. This area is located downstream of the Rahad Irrigation Scheme of Sudan established in 1981. The waterlogging and woodland areas occurred in 1998 and 2011 resulted from the drainage water of the project accumulated over the years (i.e. clear example of LULC multi-temporal change over the Rahad basin). The lower maps show the Google Earth imagines of the large-scale area. Although these Google Earth imagines dates do not exactly match the ones of the satellite images, they show the part of the dried period in the study area and hence the complexity of the LULC patterns"*

[Figure]

Figure 3_modified: Classified LULC maps of the years 1972, 1986, 1998 and 2011.

[Figure]

**Figure 4new**: Classified LULC maps of the years 1972, 1986, 1998 and 2011. The areas in the red boxes showed in large-scale to provide more details of the LULC patterns.

2. Hydrological Modelling:

**Comment:** Figure 1. I recommend to insert the Blue Nile River in the upper right map.

**Response:**

Agreed, Blue Nile River was inserted in the upper right map. Also, the rain gauges were added to this Figure (see Figure1_modified).

[Figure]

Figure 1_modified: Location map of the Dinder and Rahad basins and the DNP. The two black stars show the locations of the hydrological stations (Al-Gewisi and Al-Hawata), and the red triangles show the locations of the rain gauges.

**Comment:** Page 5, line 15: Why did you reclassify the 44 soil mapping units into 8 dominant soil groups? Was this necessary for the hydrological modelling?

**Response:**

Yes, this was necessary to comply with parametrization of the Wflow hydrological model. The reclassification of the 44 soil units into 8 dominant soil groups reduces the number of estimated parameters to 64 instead of 352. We have added this explanation in text on page 5, section 3.1.1.

*"This was necessary to reduce the model complexity. The WFlow soil model requires estimates of 8 parameters per soil type, which means 352 parameters if it is for 44 soil types. Therefore, reclassification of soil map into 8 dominant soil groups reduces the number of estimated parameters to 64".*

**Comment:** Figure 2: The WFlow_sbm model needs more explanation. How is runoff generation modelled? How is ETA calculated? Is there no interflow component?

**Response:**

We have added below text to further describe the Wflow model.
*"Combination of the total rainfall and evaporation under condition of saturated canopy is done for each rainfall storm to determine average values of precipitation and evaporation from the wet canopy. In case the soil surface is partially saturated, the rainfall that falls on the saturated area is directly added to the surface runoff component. The soil is represented by a simple bucket model that assumes an exponential decay of the saturated conductivity with depth. Lateral subsurface flow is simulated using the Darcy equation. Soil depth is identified for different land use types and consequently scaled using the Topographic Wetness Index. The WFlow_sbm interception and soil model's equations are presented in Appendix A." Further details of the Wflow model are also given at* https://media.readthedocs.org/pdf/wflow/latest/wflow.pdf.

**Comment:** Chapter 3.4: The IHA approach should be explained in more detail (add app. half page)

**Response:**

We have added below text to further describe the IHA approach.

*"The IHA method computes 33 hydrologic parameters for each year. For analyzing the alteration between two periods, the IHA described in Richter et al. (1996) was applied using the IHA software developed by The Nature of Conservancy (2009). The general approach is to define hydrologic parameters that characterized the intra-annual variation in the water system condition and then to use the analysis of variations in these parameters as a base for comparing hydrologic alterations of the system before and after the system has been altered by various human activities. The IHA method has four steps: a) define the time series of the hydrologic variable (e.g. streamflow) for the two periods to be compared; b) calculate values for hydrologic parameters; c) compute intra-annual statistics; and d) calculate values of the IHA by comparing the intra-annual variation before and after the system has been altered and present the results as a percentage of deviation. For assessing hydrologic alteration in the Dinder and Rahad Rivers, the flows variations for both rivers have been characterized based on the variations in the streamflow characteristics between three periods (1972-1986), (1986-1998) and (1998-2011). Temporal variability of streamflow series was analyzed at Al-Gewisi station on the Dinder River and at Al-Hawata station on the Rahad River."*

**Comment:** Table 2: Please explain the accuracy assessment. What means "producer" and "user"?
Did you perform a cross validation analysis? The accuracy seems very high with little uncertainty for all classes. Can you prove this?

**Response:**

The accuracy assessment is based on comparing reference data (class types at specific locations from ground information) to image classification results at the same locations. The overall accuracy of classification is the average value from all classes. The user's accuracy corresponds to errors of inclusion (commission errors), which represents the probability of a pixel

classified into a given category actually represents that category on the ground (i.e. from the perspective of the user of the classified map, how accurate is the map?). The producer's accuracy corresponds to errors of exclusion (omission errors), which represents how well reference pixels of the ground cover type are classified (i.e. from the perspective of the maker of the classified map, how accurate is the map?).

The cross validation for the year 2011 land use was made using the reference data (120 points) collected with GPS instrument during the field survey (2011–2013) assuming no significant change during this period. In addition, visual interpretation and historical information obtained from the local indigenous about the land use types in the study area were used also as cross-check validations for old maps (please refer to the text on page 6, line 17 to page 7, line 2).

Below is an example of the classification error matrix for the image of the year 2011. The error of commission and omission varies from 75 to 94% for the different LULC classes.

| Image Date: 2011 | | Class types determined from reference data | | | | | | |
|---|---|---|---|---|---|---|---|---|
| Class types determined from classified map | LULC | Woodland | Shrub Land | Grass Land | Crop Land | Bare Land | Water | Row Total |
| | Woodland | 130 | 3 | 2 | 2 | 1 | 2 | 140 |
| | Shrub Land | 4 | 36 | 4 | 4 | 0 | 0 | 48 |
| | Grass Land | 1 | 2 | 68 | 2 | 3 | 0 | 76 |
| | Crop Land | 2 | 3 | 3 | 45 | 2 | 0 | 55 |
| | Bare Land | 1 | 1 | 2 | 1 | 28 | 0 | 33 |
| | Water | 5 | 0 | 0 | 0 | 0 | 30 | 35 |
| | Column Total | 143 | 45 | 79 | 54 | 34 | 32 | 387 |
| Accuracies (%) | | | | | | | | |
| Producer's | | 91* | 80 | 86 | 83 | 82 | 94 | |
| User's | | 93** | 75 | 89 | 82 | 85 | 86 | |
| Overall | | 87 | | | | | | |

\* 91% = (130/143) x100
\*\* 93% = (130/140) x 100

We have added the following text to the manuscript in section 4.1 to give further clarification on accuracy assessment.

*"The accuracy assessment is based on comparing reference data (class types at specific locations from ground information) to image classification at the same locations. The overall accuracy of classification is the average value from all classes. The user's accuracy corresponds to errors of inclusion (commission errors), which represents the probability of a pixel classified into a given class actually represents that class on the ground (i.e. from the perspective of the user of the classified map). The producer's accuracy corresponds to errors of exclusion (omission errors), which represents how well reference pixels of the land cover type are classified (i.e. from the perspective of the maker of the classified map). The commission errors occur when an area is included in an incorrect category, while the omission errors occur when an area is excluded from the category to which it belongs. Every error on the map is an omission from the correct class and a commission to an incorrect class (Congalton and Green 2008). The cross validation for the year 2011 land use was made using the reference data (120 points) collected with GPS instrument during the field survey (2011–2013). In addition, visual interpretation and historical information obtained from the local people about the land use types in the study area were used also as cross-check validations for old maps. Shrublands show lower user's and producer's accuracies compared to the other LULC classes. This is mainly due to the miss-classification of some shrub land into woodland, grassland and cropland."*

**Comment:** Chapter 4.1.1 / Figure 4 and Figure 5: A critical discussion about the calibration and its uncertainties is totally missing. Couldn't you assess the reliability of the RS data by ground truth measurements (rain gauges)? Please comment on that. Concerning the figures, there are great differences in the peak flows with reverse biases. For instance, at Al-Gewisi station, you get a large underestimation in the first validation period for CHIRPS, whereas you get a large underestimation for the same time period and RS method for the Al-Hawata station. There are many contrary results comparing the two figures. Please

discuss this issue. Moreover, did you vary the plant parameters for different crops (Root depths, crop coefficient, LAI, etc.)? Please name and quantify the parameters.

**Response:**

Our response to this comment is divided into three parts as follows:

**Comment:** Chapter 4.1.1 / Figure 4 and Figure 5: A critical discussion about the calibration and its uncertainties is totally missing. Couldn't you assess the reliability of the RS data by ground truth measurements (rain gauges)? Please comment on that.

**Response:**

We must admit that we haven't conducted an in-depth validation of the Satellite Based Rainfall Estimate (SBRE) products. It is beyond the scope of this manuscript. However, we assessed the reliability of SBRE products by direct comparison against ground measurements at four locations. The results are shown in Figure 5new.

We have inserted below text including Figure 5 new in section 4.1.1 of the manuscript

*"To assess the reliability of the SBRE products, validation is done with the use of ground measurements at four gauges in which observed data are available. Two gauges (Gonder and Bahir Dar) are located nearby the upstream part of the catchments in the Ethiopian plateau, while the others two (Gedarif and Al-Hawata) are located at the most downstream part of the catchment in the Sudan low land. The validation is performed at annual time step. The results show that the difference of RFE against ground measurements has no consistent patterns. TRMM and CHIRPS have shown no consistent patterns at the lowland (Gedarif and Al-Hawata), but both products are consistent and overestimate rainfall at the Ethiopian highland (Gonder and Bahir Dar) in all years except 2007 (Figure 5). Since both the Dinder and Rahad derive their main flow from the Ethiopian highlands, products with consistent patterns at highlands will be more suitable for running hydrologic models in this catchment. From these findings, one can conclude that the CHIRPS v2.0 and TRMM 3B42 v7 are more suitable than RFE 2.0 for running hydrologic model. Comparing CHIRPS v2.0 and TRMM 3B42 v7, it is clear that CHIRPS v2.0 has less overestimation of rainfall. Thus, CHIRPS v2.0 is the best product to be used as a forcing data for hydrologic model in the Dinder and Rahad Basins".*

[Figure]

[Figure]

[Figure]

Figure 5new: Comparison of SBRE products with ground measurements at four locations

**Comment:** Concerning the figures, there are great differences in the peak flows with reverse biases. For instance, at Al-Gewisi station, you get a large underestimation in the first validation period for CHIRPS, whereas you get a large underestimation for the same time period and RS method for the Al-Hawata station. There are many contrary results comparing the two figures. Please discuss this issue.

**Response:**

The differences in the peak flow with reverse bias can be attributed to the differences in rainfall estimation with different products (see Figure 5new). At Al-Gewisi station, the large underestimation in the first validation period for CHIRPS can be attributed to the underestimation of rainfall by CHIRPS in 2007 at both Gonder and Bahir Dar (Figure 5new). At the same time CHIRPS overestimates rainfall in all years from 2001 to 2006. Therefore, calibration of the hydrologic model based on (2002-2005) will result in a more underestimation of river flow for the year 2007. This is confirmed by Figure 4 in the original manuscript and supported by the validation result in Figure 4new presented in this document. On the other hand, at Al-Hawata station the difference between observed and model flow in the first period of validation (2008) is difficult to explain in terms of problem in model structure as it is a single event out of the 4 years validation period. It can likely be due to error either in the input data or the observed flow or a combination of both. Results of the three SBRE products at Al-Gewisi and Al-Hawata (Figure 6 and 7) are consistent and supported by the validation results in Figure 5new. For instance, in 2004 the model results for RFE 2.0 underestimate the flow at both Al-Gewisi and Al-Hawata stations. This is supported by the rainfall validation results in Figure 5new which shows that RFE 2.0 underestimates rainfall in all stations. Also in 2010, the model results for TRMM 3B42v7 underestimate the flow at both Al-Gewisi and Al-Hawata stations. This is also supported by the validation results in Figure 4new which shows that TRMM 3B42v7 underestimates rainfall in Gedarif station. The text hereafter is added to section 4.1.1.

*"At Al-Gewisi station, the large underestimation in the first validation period for CHIRPS can be attributed to the underestimation of rainfall by CHIRPS in 2007 at both Gonder and Bahir Dar (see Figure 5). At the same time CHIRPS overestimates rainfall in all years from 2001 to 2006. Therefore, calibration of the hydrologic model based on (2002-2005) resulted in a more underestimation of river flow in 2007. On the other hand, at Al-Hawata station, the difference between observed and model flow in the first period of validation (i.e.2008) is likely due to an error either in the input data or the observed flow or a combination of both."*

**Comment:** Moreover, did you vary the plant parameters for different crops (Root depths, crop coefficient, LAI, etc.)? Please name and quantify the parameters.

**Response:**

Yes, indeed each vegetation cover is assigned different parameters when modeling the flow response to different land cover. These parameters include; root depth, leaf area index (LAI), evaporation from wet canopy/average rainfall (E/R) ratio, Albedo, Canopy Gap Fraction and Maximum Canopy Storage. All model parameters are linked to the Wflow model through lookup tables. The text below is added to the manuscript.

*"Different parameters are assigned to each land cover type. These parameters include; rooting depth, leaf area index (LAI), ratio of evaporation from wet canopy to average rainfall ($E_w/R$), Albedo, Canopy Gap Fraction and Maximum Canopy Storage. All model parameters are linked to the Wflow model through lookup tables. The lookup tables are used by the model to create input parameter maps. Each table consists of four columns. The first column is used to identify the land-use class, the second column indicates the sub catchment, the third column represents the soil type and the last column lists the assigned values based on the first three columns. The parameters are linked to land use, soil type or sub-catchment through lookup tables. Description of the Wflow model parameters is presented in Appendix B and the calibrated values for each parameter are presented in Appendix C."*

**Comment:** Chapter 4.2: You should not only the resulting streamflow pattern (Figure 6) but also the different ETA – for same HRUs (Hydrological response units) and for the entire catchment. How was the water balance changed?

**Response:**
In Figure 6 (Figure 8 in the revised marked-up manuscript), we have shown the effect of LULCC on the daily streamflow (peak flows). However, to show the changes in AET at HRUs (sub-catchments) and the water balance for the entire catchment, the discussion in the text and the new tables (Table 4 and 5) below are added to section 4.2.

*"In addition to the streamflow response to LULCC, Evapotranspiration (ET) is another important component of the water balance that constitutes a major determinant of the amounts of water draining from different land cover type within the catchment. The ET result shows high rates of actual evapotranspiration (AET) when running the model with land cover of 1972 and 1998 at both the sub-catchments and the entire catchment (Table 4 and 5). This can be attributed to the large percentage coverage of woodland in 1972 and 1998 compared to land cover of 1986 and 2011 (please refer to table 3 in the main manuscript). The lowest AET is observed when running the model with land cover of 1986. This is likely due to the severe drought during the mid-1980s that limits the water availability and decreases the green coverage. Table 4 presents the changing in the annual average AET at sub-catchment level as a response to LULCC for the Dinder catchment. Table 5 shows the changes in water balance for the entire Dinder and Rahad catchments when running the hydrologic model with different LULC and fixed rainfall data for the periods (2001-2012)."*

Table 4: Annual average AET as a response to LULCC at the sub-catchments level for the Dinder catchment (1972-1986).

| Year | AET from land cover of 1972 (mm) | | | | AET from land cover of 1986 (mm) | | | |
|------|-----------|------|--------|--------------|-----------|------|--------|--------------|
| | Al-Gewisi | Musa | Gelagu | Upper Dinder | Al-Gewisi | Musa | Gelagu | Upper Dinder |
| **2001** | 558 | 583 | 626 | 464 | 426 | 424 | 396 | 288 |
| **2002** | 443 | 456 | 535 | 510 | 322 | 317 | 306 | 312 |
| **2003** | 564 | 639 | 642 | 486 | 425 | 469 | 405 | 312 |
| **2004** | 455 | 502 | 573 | 500 | 326 | 354 | 340 | 311 |
| **2005** | 504 | 547 | 575 | 505 | 376 | 396 | 358 | 323 |
| **2006** | 527 | 576 | 632 | 545 | 396 | 414 | 406 | 359 |
| **2007** | 598 | 602 | 618 | 564 | 468 | 444 | 400 | 382 |
| **2008** | 593 | 689 | 703 | 576 | 459 | 513 | 471 | 392 |
| **2009** | 421 | 482 | 519 | 516 | 310 | 343 | 302 | 323 |
| **2010** | 536 | 566 | 606 | 520 | 412 | 415 | 383 | 331 |
| **2011** | 470 | 467 | 554 | 530 | 350 | 327 | 329 | 332 |
| **2012** | 636 | 679 | 684 | 542 | 500 | 504 | 450 | 353 |

Table 5:   Water balance of the Dinder and Rahad catchments as a response to LULCC

| Dinder catchment | Land cover of 1972 | | Land cover of 1986 | | Land cover of 1998 | | Land cover of 2011 | |
|---|---|---|---|---|---|---|---|---|
| Year | Rainfall (mm) | AET (mm) | Streamflow (mm) | AET (mm) | Streamflow (mm) | AET (mm) | Streamflow (mm) | AET (mm) | Streamflow (mm) |
| 2001 | 816 | 558 | 258 | 383 | 433 | 432 | 384 | 496 | 320 |
| 2002 | 663 | 486 | 177 | 314 | 349 | 364 | 299 | 430 | 233 |
| 2003 | 847 | 583 | 264 | 403 | 444 | 449 | 397 | 519 | 327 |
| 2004 | 703 | 507 | 195 | 333 | 370 | 374 | 329 | 451 | 252 |
| 2005 | 768 | 532 | 236 | 363 | 405 | 414 | 354 | 479 | 289 |
| 2006 | 835 | 570 | 265 | 394 | 441 | 441 | 395 | 513 | 322 |
| 2007 | 876 | 595 | 280 | 424 | 452 | 476 | 400 | 540 | 336 |
| 2008 | 929 | 640 | 289 | 459 | 470 | 509 | 420 | 582 | 347 |
| 2009 | 659 | 484 | 175 | 319 | 340 | 363 | 297 | 435 | 225 |
| 2010 | 817 | 557 | 260 | 385 | 432 | 432 | 386 | 505 | 312 |
| 2011 | 710 | 505 | 205 | 334 | 376 | 377 | 333 | 454 | 256 |
| 2012 | 972 | 635 | 337 | 452 | 520 | 498 | 474 | 579 | 393 |
| **Rahad catchment** | Land cover of 1972 | | Land cover of 1986 | | Land cover of 1998 | | Land cover of 2011 | |
| Year | Rainfall (mm) | AET (mm) | Streamflow (mm) | AET (mm) | Streamflow (mm) | AET (mm) | Streamflow (mm) | AET (mm) | Streamflow (mm) |
| 2001 | 724 | 409 | 315 | 290 | 434 | 398 | 326 | 309 | 416 |
| 2002 | 641 | 398 | 243 | 271 | 370 | 383 | 258 | 291 | 350 |
| 2003 | 755 | 450 | 305 | 323 | 432 | 434 | 322 | 342 | 413 |
| 2004 | 609 | 360 | 249 | 231 | 378 | 338 | 270 | 244 | 364 |
| 2005 | 656 | 399 | 258 | 267 | 389 | 378 | 278 | 285 | 372 |
| 2006 | 782 | 450 | 332 | 324 | 457 | 431 | 351 | 336 | 446 |
| 2007 | 774 | 473 | 301 | 344 | 430 | 456 | 319 | 363 | 411 |
| 2008 | 754 | 438 | 315 | 313 | 441 | 415 | 338 | 322 | 431 |
| 2009 | 581 | 352 | 229 | 220 | 361 | 333 | 248 | 238 | 343 |
| 2010 | 744 | 449 | 295 | 319 | 425 | 431 | 313 | 335 | 409 |
| 2011 | 610 | 369 | 241 | 235 | 375 | 348 | 262 | 252 | 358 |
| 2012 | 873 | 507 | 366 | 381 | 492 | 485 | 388 | 390 | 483 |

**Comment:** Page 17, line 22: "In the Dinder River the effect of LULCC on streamflow is not big as in Rahad River." Please find reasons for this different behavior.

**Response:**

The reason for this was mentioned in the conclusion on page 24, lines 2-4 in the revised marked-up version of the manuscript. However, for more clarity the explanation in the text below is added to section 4.3.2, Page 22, lines 3-4.

*"This is likely due to the large expansion in cropland in the Rahad catchment to 68% of the total area compared to 47% in the Dinder catchment".*

**Comment:** Conclusions: Please analyze also the effect of different precipitation patterns and magnitudes on streamflow in different years (2001 until 2012). When do you see a larger effect of LULCC on streamflow alteration? Find explanations for that.

**Response:**

Replying to the first part of this comment "Please analyze also the effect of different precipitation patterns and magnitudes on streamflow in different years (2001 until 2012)", the text below in addition to the new figure is added to section 4.2.

*"Since both Dinder and Rahad rivers are seasonal, their flows are mainly depending on rainfall patterns and magnitudes. In addition to the effect of LULCC on the streamflow, Figure 10(new) shows that the annual variability of rainfall is another factor affecting the annual patterns of the streamflow." Since Figure 10new is a new Figure, figures in the manuscript are updated accordingly."*

[Figure]

Figure 10: Annual average rainfall and streamflow patterns and magnitudes for the years (2001-2012).

**Response:**

Our reply to the second part of the above comment "When do you see a larger effect of LULCC on streamflow alteration? Find explanations for that.", is as follows:

The effect of LULCC on streamflow is found to be large when running the model with land use of 1986 and 2011 particularly in Rahad River. This could be attributed to the severe drought during 1984/1985 that accelerates the runoff processes due to land degradation, beside the large expansion in cropland in the Rahad catchment to 68% of the total area in 2011 compared to 47% in the Dinder catchment. This has been mentioned in the conclusion section on page 24 lines 2-4 (Please refer also to Revision 2.19).

[revised manuscript text omitted]